evolution, behaviour

acoustic communication, signal, phenotypic plasticity, peak shift, adaptive landscape, Passeriformes

**Author for correspondence:**
Jay P. McEntee
e-mail: jaymcentee@missouristate.edu

†Present address: Department of Biology, Missouri State University, 901 S. National Avenue, Springfield, MO 65897, USA.

# Punctuated evolution in the learned songs of African sunbirds

Jay P. McEntee[1,2,†], Gleb Zhelezov[3], Chacha Werema[4], Nadje Najar[5], Joshua V. Peñalba[6], Elia Mulungu[7], Maneno Mbilinyi[8], Sylvester Karimi[9], Lyubov Chumakova[3], J. Gordon Burleigh[10] and Rauri C. K. Bowie[1]

[1]Museum of Vertebrate Zoology and Department of Integrative Biology, University of California, Berkeley, CA 94720, USA
[2]Department of Biology, Missouri State University, Springfield, MO 65897, USA
[3]School of Mathematics, University of Edinburgh, Edinburgh EH9 3FD, UK
[4]Department of Zoology and Wildlife Conservation, University of Dar-es-salaam, PO Box 35064, Tanzania
[5]School of Natural Resources, University of Nebraska, Lincoln, NE 68503, USA
[6]Museum für Naturkunde, Center for Integrative Biodiversity Discovery, Invalidenstrasse 43, 10115 Berlin, Germany
[7]PO Box 934, Iringa, Tanzania
[8]Tanzania Bird Atlas, Iringa, Tanzania
[9]National Museums Kenya, Nairobi, Kenya
[10]Biology Department, University of Florida, Gainesville, FL 32611, USA

JPM, 0000-0002-1213-9734; JVP, 0000-0001-6549-6885

Learned traits are thought to be subject to different evolutionary dynamics than other phenotypes, but their evolutionary tempo and mode has received little attention. Learned bird song has been thought to be subject to rapid and constant evolution. However, we know little about the evolutionary modes of learned song divergence over long timescales. Here, we provide evidence that aspects of the territorial songs of Eastern Afromontane sky island sunbirds *Cinnyris* evolve in a punctuated fashion, with periods of stasis of the order of hundreds of thousands of years or more, broken up by evolutionary pulses. Stasis in learned songs is inconsistent with learned traits being subject to constant or frequent change, as would be expected if selection does not constrain song phenotypes over evolutionary timescales. Learned song may instead follow a process resembling peak shifts on adaptive landscapes. While much research has focused on the potential for rapid evolution in bird song, our results suggest that selection can tightly constrain the evolution of learned songs over long timescales. More broadly, these results demonstrate that some aspects of highly variable, plastic traits can exhibit punctuated evolution, with stasis over long time periods.

## 1. Background

Signal evolution has long been thought to be important to the process of animal speciation [1], in part because many closely related species have distinct signals while differing little in other traits [2,3]. In particular, the evolution of signals involved in mate choice has been thought to be critical to the evolution of pre-mating reproductive isolation [1], such that correlated evolution of signals and mating preferences could lead in and of itself to speciation [4,5]. However, there remain many questions about how signal divergence proceeds over time, which mechanisms are responsible, and how it contributes to speciation and diversification processes.

Some signals that may be important to speciation are highly plastic, including those that are impacted by learning processes [6–8]. While divergence in less plastic traits generally requires genetic divergence, the same is not true for learned signals (even if they have components with genetic predispositions [9,10]). Indeed, novel learned signals can arise without genetic mutation, and spread quickly throughout populations [11]. Such change could serve as an

initial step in divergence that includes subsequent genetic changes. Thus, learned signals are subject to different evolutionary pressures, including the introduction of culturally transmitted novelties and cultural drift [2,7], and may exhibit different evolutionary rates [12] or trajectories (e.g. extents of gradualism versus punctuated evolution) over time [13], when compared with signals that are not learned. Learned signals may be especially subject to regular and rapid evolutionary change because cultural novelties appear frequently relative to genetic mutations [2,14].

The songs of oscine songbirds present intriguing cases for the study of learned signal evolution. Most oscine songbirds learn to perform aspects of songs by imitating conspecifics [15]. The oscine learning process is directed by innate predispositions that result in selective learning—that is, oscine individuals only learn or reproduce vocalizations with certain characteristics typical of their species [16–18]. Thus, diversification of oscine song may involve cultural evolution, the evolution of the innate predispositions, and potentially the interactions between these levels.

In this study, we are interested in assessing the tempo and mode of learned bird song evolution. Previous reviews [2,13] have suggested that learned signals should exhibit little conservatism, with isolated populations typically evolving different phenotypes via cultural evolution even before any genetic differences have accrued. If rapid cultural evolution serves as a first step in further divergence that includes underlying genetic change, learned songs may diverge gradually over time, and potentially at high rates. Gradual divergence in song may also occur if the innate predispositions that guide song development are themselves not strongly constrained, and are free to diverge by genetic drift or novel selective pressures. Indeed, gradual evolution has been posited to be important in song divergence connected to speciation [19]. The strongest empirical evidence for the evolutionary trajectory of learned song relevant to speciation likely comes from studies of the greenish warbler *Phylloscopus trochiloides*, which exhibits nearly continuous variation across geographical space, suggestive of gradual evolution [20].

However, it remains possible that learned song can exhibit punctuated evolution, where pulses of divergence occur against a backdrop of stasis, or highly bounded, non-accumulating evolution. Such a pattern may arise if innate song-learning predispositions remain fixed for long periods of time, with infrequent pulses of evolutionary change. Modelling suggests that a build-up of genetic variation in song-learning predispositions can occur despite constancy in song phenotypes, potentially providing the means for rapid, pulse-like changes following periods of stasis [12]. Interestingly, prior empirical studies have documented that aspects of song can be highly similar across broad geographical ranges [21] and over hundreds of years [22–24], suggesting that longer-term stasis in spectrotemporal characters is feasible. However, for species that are similar across broad geographical ranges, gene flow or recent range expansion could account for similarity, so it is unclear whether these examples exemplify the potential for conservatism over longer timescales. Recent empirical evidence from white-throated sparrows *Zonotrichia albicollis* showing apparent rapid replacement of one song form by another may correspond well to a punctuated evolution model. However, it is unclear for how long the prior variant dominated. If such pulses occur regularly, this process would appear as gradual evolution over

longer timescales, whereas if such pulses occur rarely against a backdrop of stasis, this process would appear as punctuated evolution. What is needed to understand whether bird songs evolve gradually or via more punctuated evolution are studies that allow assessment of song divergence across a spectrum of molecular divergence levels, capturing earlier and later stages of divergence.

To accomplish this, we investigated the learned territorial/mate-attracting songs (hereafter, territorial songs) of the eastern double-collared sunbird (EDCS) species complex [25], which inhabits mountains of the Eastern Afromontane. Sunbirds are oscine songbirds (the largest clade of song-learning birds, comprising approx. 5000 species [26]), and their territorial songs exhibit signatures of songs developed through learning, including striking complexity and variation [15]. The geographical ranges of these species are archipelago-like (figure 1), with populations occupying discrete, island-like patches of suitable montane forest and forest edge habitats. Similar to many other co-distributed taxa in the Eastern Afromontane and especially the Eastern Arc Mountains, the EDCS complex exhibits deep genetic structure across sky islands, both among and within named taxa, indicating that many populations have experienced long periods of isolation. We recorded territorial songs across this distribution, then quantified song variation across hierarchical levels of organization. We then used discriminant analyses to assess whether clustering of songs in multivariate space corresponded with molecular lineages that have been characterized as different species. Then we examined the evolutionary mode (i.e. gradualism versus punctuated evolution) of learned songs across the species complex. We fit models representing different temporal evolutionary trajectories to song via phylogenetic methods, and test which model better fits the evolution of song traits. This question has rarely been posed for signalling phenotypes [27].

## 2. Results

We used linear mixed-effect models to assess variation in 14 song traits across the measured songs in our dataset. Across 356 songs from 123 individuals, the random-effects variance terms in these models indicated that there was substantial variation within and among individuals, but little geographical variation within species. The variance terms corresponding to variance among individuals (within populations) and the residual variance (including within-individual variation) accounted for greater than 89% of the random-effects variance across all traits. The among-population, within-species component contributed less than 11% of the random-effects variance for all 14 measured song traits (electronic supplementary material, table SA1), indicating that geographical variation within species is minor relative to within-population variation. The proportion of the variance explained by the fixed effect varied from 0.07 to 0.82 across traits, with a mean of 0.49, indicating that among-species differences were substantial across most of the 14 traits. We note that for this analysis, we divided *Cinnyris mediocris* into northern and southern groups (figure 1), creating an additional level of the fixed effect because these groups differed strongly in some song traits. For the models presented in the main text, we excluded the two populations found at the centre of the *Cinnyris fuelleborni–Cinnyris moreaui* hybrid zone, as assigning

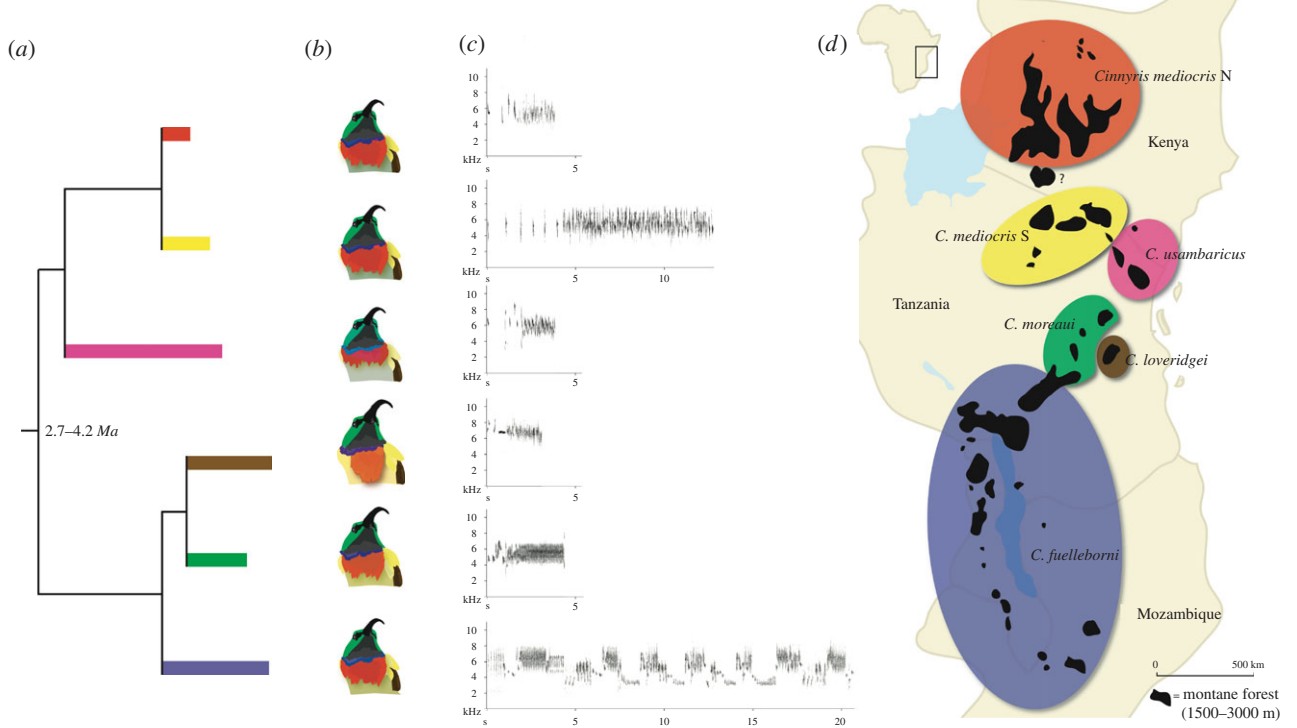

**Figure 1.** An overview of the EDCS species complex. (*a*) A phylogenetic tree trimmed to include named species and one within-species division that corresponds with a major song divergence. Estimated age of the MRCA is shown at the node. (*b*) Depictions of typical adult male plumage for the six lineages represented. (*c*) Sonograms showing representative songs for the six lineages shown. (*d*) Ranges of the six lineages in eastern Africa across Kenya, Tanzania, Malawi and Mozambique. (Online version in colour.)

individuals to species in these populations is more challenging, and some song convergence occurs [28]. The results of linear mixed modelling were similar; however, when including the two populations at the centre of the hybrid zone minus a single *C. moreaui* with especially aberrant song from this area (*n* = 141, electronic supplementary material, table SA2).

We additionally performed model-based discriminant analyses of song phenotypes at the individual level (means across songs for individuals, *n* = 123 individuals) to confirm that song phenotypes formed clusters in multivariate space that corresponded to species. Song phenotypes clustered by species, with additional separation between southern and northern populations of *C. mediocris* (electronic supplementary material, figure SA1). Classification error to the six groups shown in figure 1 was 0%. When performing the same analysis including populations at the centre of the *moreaui/fuelleborni* hybrid zone (*n* = 141 individuals, electronic supplementary material, figure SA2), classification error was 0.7%.

Multi-locus phylogenetic analyses based on the mtDNA gene ND2 and five nuclear intron sequences revealed molecular lineages that correspond with song phenotype clusters (figure 1; electronic supplementary material, figures SA2 and SA3). We recovered five major molecular lineages across the species complex that were similar to those found in previous phylogenetic analyses using only mtDNA sequences [25], and correspond with the taxonomy proposed in that study. Additionally, we recovered distinct clades representing isolated populations within three species: *C. mediocris*, *C. fuelleborni* and *C. moreaui*. In *C. mediocris*, our samples from the Mbulu highlands in northern Tanzania formed a clade, while those from Kenyan populations formed a clade sister to it. *Cinnyris fuelleborni* also comprised two clades,

with individuals from the Njesi Plateau in northern Mozambique sister to all other *C. fuelleborni*. In *C. moreaui*, samples from the Nguru Mountains formed a clade nested within a phylogenetic grade representing samples from all other localities for this taxon (electronic supplementary material, figure SA3). Phylogenetic analysis at the species level using BEAST recovered the same topology for species relationships for the five named species as our ML analysis, and estimated a divergence time of 3.4 My (HPD interval: 2.67–4.18 My) for the most recent common ancestor (MRCA) of the EDCS species complex (electronic supplementary material, figure SA4). Our population trees, which we used to fit evolutionary models for song phenotypes (e.g. figure 2), recovered the same topology among named taxa as the ML and Bayesian analyses.

## (a) Inference of tempo and mode of learned song evolution

Using novel model-fitting approaches and a novel model implementation for punctuated evolution on phylogenetic trees (see electronic supplementary material), we compared support for Brownian motion versus punctuated evolution across different song traits. We found strong support for punctuated evolution in four of the 14 song traits (log duration, range of peak frequency, CV peak frequency, median peak frequency) and moderate support for punctuated evolution in three more traits (median pause duration, median element duration, log number of elements). For the remainder (CV frequency bandwidth, CV frequency change, maximum peak frequency, minimum peak frequency, log median frequency change, CV pause duration, log median bandwidth), neither model was strongly favoured over the other. Sensitivity analyses examining the fit of trait evolution models on bootstrap

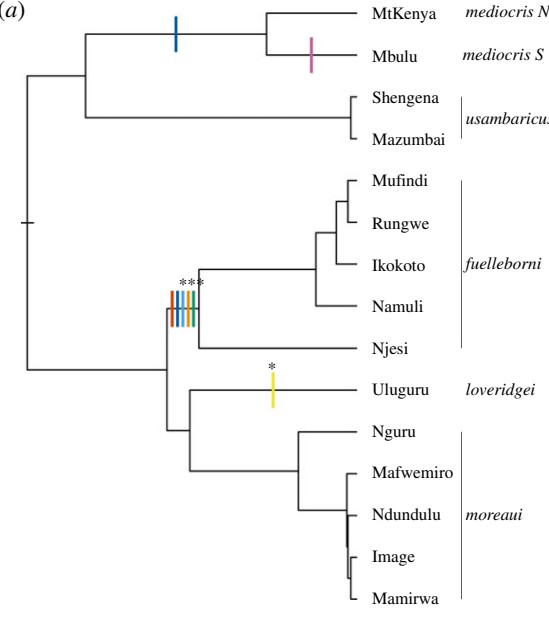

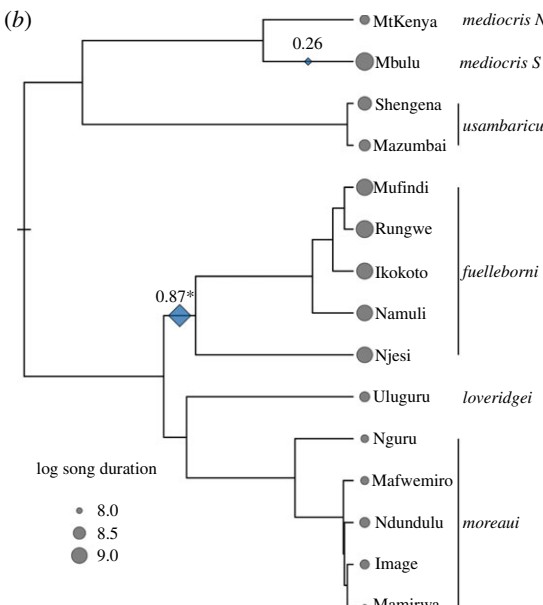

**Figure 2.** Pulse localization. (*a*) Pulse localization for all song traits with moderate or strong support for pulsed evolution. The pulses shown are only those that occurred in the preferred pulse configuration as determined by AICc. Those pulses corresponding to traits with strong support following our simulation-based correction are denoted with an asterisk. The others come from traits with moderate support (see text). Colours represent traits as follows: green, log song duration; orange, range peak frequency; light blue, CV peak frequency; yellow, median peak frequency; dark blue, median pause duration; red, median element duration; pink, log number of elements. The tips of the phylogenetic tree correspond to populations. Species epithets are indicated at far right. (*b*) Support for localization based on AICc weighting among pulse configurations for the evolution of (log) song duration. Blue diamonds are found on branches where pulse localizations had support values greater than 0.2, with diamond size reflecting support, and support value shown above. The asterisk signifies a pulse that occurred in the pulse configuration with the minimum AICc value. Sizes of grey circles correspond to mean phenotype values at tips, which each represent a geographically discrete sky island population. Species epithets are indicated at far right. This phylogenetic tree was constructed based on genetic distances among populations for mtDNA genes, and corresponds in topology among species to trees built using maximum likelihood and Bayesian approaches from multi-locus datasets. (Online version in colour.)

trees showed that our results were broadly robust to phylogenetic uncertainty (electronic supplementary material, figure S4).

Our approach allowed us to compare support for the number and positions of evolutionary pulses on our phylogenetic tree. Pulses were allowed to occur on any branch. In our modelling approach, each pulse was considered a parameter, such that more parsimonious models had fewer pulses. We present the results for our pulse localization approach for all song traits where pulsed evolution had strong or moderate support (an example is shown in figure 2*b*; see also electronic supplementary material, figures SA4–SA9 and table S2). Pulses for four of these seven traits were co-localized to the branch representing the common ancestor of *C. fuelleborni* populations, with support values greater than 0.8. The pulse configurations with minimum AICc for the seven traits where punctuated evolution had moderate or strong support had a mean of $1.14 \pm 0.35$ s.d. pulse positions (electronic supplementary material, figure S8). Support for the punctuated evolution model across these traits, coupled with the small number of pulses supported for them, is consistent with the hypothesis that evolutionary change is minimal for these traits for long stretches of time, corresponding to hundreds of thousands of years or more on the phylogenetic tree.

## 3. Discussion

### (a) Learned song evolution as peak shifts on adaptive landscapes

Here, we have presented evidence that punctuated evolution explains the evolution of aspects of territorial song better than gradual evolution (Brownian motion), across the EDCS species complex. In other song traits, we find equivocal support for punctuated and gradual evolution. Both temporal and spectral aspects of song were among those with strong support for punctuated evolution. Our results support the notion that aspects of learned songs can evolve by large jumps amid extended periods of highly bounded evolution (in which short-term evolution is non-accumulating over time), or stasis. While rapid evolution of animal signals has often been discussed in the literature on signal evolution, and has frequently been invoked as a route to pre-zygotic reproductive isolation, extended periods of stasis in signals has received comparatively little attention [27]. Characterizing the evolutionary mode of territorial song as we have done here sheds light on the form of signal evolution, how it may be involved in speciation processes, and what may or may not cause evolutionary changes.

We focus largely on the support for punctuated evolution here; however, there was variation in relative support for punctuated versus gradual evolution across traits. Support for punctuated evolution may be associated with the extent of differences among species for a given trait. The variance explained by species identity (marginal $R^2$ from LMMs; electronic supplementary material, table SA1) was higher for traits with strong or moderate support for punctuated evolution ($0.58 \pm 0.07 = 8$, $n = 7$), than for traits with equivocal support for punctuated and gradual evolution ($0.40 \pm 0.07$, $n = 7$). This evidence suggests that song traits that evolve more gradually may not achieve levels of difference that distinguish taxa as often as those traits that evolve by pulses.

The learning process in the development of song in oscine songbirds, like the focal sunbirds here, is a form of phenotypic plasticity [7]. As in previous studies using phylogenetic comparative methods [29,30], we interpret punctuated evolution by visualizing peak shifting on an adaptive landscape. The two-dimensional version of the adaptive landscape we visualize is defined by having a trait on the x-axis, and the y-axis representing the contribution of receiver responses to fitness [31]. Authors have suggested that phenotypic plasticity itself could assist peak shifting in adaptive landscapes by allowing phenotypes to initially approach alternate peaks without having to wait for novel genetic variation, especially in the case of behaviour [32]. However, high levels of plasticity may enable so much phenotypic variation that no underlying genetic change is required, such that adaptation to a novel peak does not occur (i.e. plasticity itself is the adaptation). In the case of song, it would seem unlikely that there is a fixed adaptive landscape [33], in which peaks maintain stable shapes and occupy the same positions through time. Instead, because the efficacy of signals can change depending on environmental variation (e.g. habitat structure [34]; population density [35]) and especially with the evolution of receiver responses [27], adaptive peaks for learned song would appear likely to change shape, move, appear and/or disappear, over time and across geographical space. Kirkpatrick [36] and Whitlock [33] showed that even small changes in the slopes and heights of adaptive peaks alone could trigger peak shifts, suggesting they may occur regularly, especially for plastic traits [32]. Thus, there are two main theoretical obstacles for highly variable learned song to exhibit peak shifting dynamics over longer timescales. First, song may be so plastic that it can travel about an adaptive landscape without any underlying genetic change [32,37], in which case it would likely be prone to extremely high lability. Secondly, the adaptive landscape for song may change rapidly through time, and vary across environments, such that adaptive peaks are unlikely to remain in the same shape and position over evolutionary timescales [27,33]. Thus, there was little reason to expect learned songs to be restricted to peaks over the timescales inferred in this study, because of their high variability and little reason to expect adaptive peaks to be stable in position and shape over time such that they could be observed.

Our finding that punctuated evolution better characterizes the evolution of some song traits than gradual evolution indicates that trait evolution can be tightly bounded, approaching stasis, over long periods of time (greater than $10^5$ years). These results suggest that adaptive peaks for song are stable over time. The stability of adaptive peaks for learned songs suggests that the songs' receivers mediate stabilizing selection on song traits. There are two sets of receivers, males and females, that are likely to exert stabilizing selection forces in sunbirds. If narrow female preferences alone were responsible for stasis, we would expect strong behavioural reproductive isolation where two species with highly divergent song come into contact. However, C. moreaui and C. fuelleborni, which have extremely different songs across many song aspects, hybridize where they come into contact [28], indicating that female preferences are unlikely to be narrow. Thus, male receivers may play an important role in this instance. They may do this by exerting stabilizing selection on their own, or by exerting directional selection opposite another force, like selection from females or viability selection. An alternate hypothesis for stasis in some traits is that

evolutionary constraints result from limited genetic variance [38–41]. However, limited genetic variance should not strongly constrain evolution over longer timescales, as examined here, because novel genetic variation will arise over these timescales.

An alternate explanation of the within-species stasis pattern observed in this study is that similarity among isolated populations has been maintained by gene flow during past periods of greater population connectivity. By this hypothesis, highly divergent molecular lineages within species in our study (Njesi within C. fuelleborni and Nguru within C. moreaui) may have experienced cryptic gene flow. Though we have not performed genome-wide sequencing for this study and, therefore, cannot completely rule out this explanation, we do not find it nearly as plausible. First, these highly divergent lineages share no mtDNA haplotypes with other populations. As mtDNA is inherited via females in birds, and as females tend to be more dispersive in passerine birds [42], this alternate explanation would likely require a mechanism for nuclear-biased gene flow without mtDNA gene flow. Secondly, there is substantial evidence for long-term isolation among sky island populations for co-distributed taxa in the Eastern Afromontane [43–45], suggesting that recent climatic changes have not generally facilitated connectivity among these sky islands.

If near-stasis occurs over long periods of time in some bird song traits, what explains evolutionary divergence when it occurs? One prominent hypothesis explaining the evolution of bird song is that song evolves as a by-product of morphological evolution. Body size evolution may be important because of allometric changes in pieces of the vocal apparatus, which could alter song frequency [46]. When song evolves by punctuated evolution, the morphological by-product hypothesis would predict that evolutionary pulses are consequences of morphological evolution (which in itself might be punctuated). In the EDCS complex, there is limited morphological evolution, with subtle changes in morphology across the complex, and substantial overlap in morphological characteristics that differ on average between species [25]. However, Loveridge's sunbird C. loveridgei is unambiguously the largest member of the species complex. From the morphological by-product hypothesis, we would predict that C. loveridgei should have the lowest frequency songs. We find the opposite. C. loveridgei sing songs with the highest peak frequencies of all the members of the species complex, and our analyses evince a pulse of peak frequency evolution unique to the Loveridge's sunbird lineage. This evidence suggests that morphological evolution does not hold the key to understanding inferred evolutionary pulses of song evolution in the EDCS complex.

Range expansion provides another possibility as a cause for pulses in learned songs. Studies on North American juncos [47,48] have suggested that pulses of phenotypic divergence (in that case, plumage) might take place in association with instances of rapid range expansion. During rapid range expansion, serial founder effects can induce the fixation of rare genetic variants, and selective forces on signals may be distinct at the leading front of range expansions. For example, population densities at the leading edges of range expansions may be low, which could advantage signals that broadcast across further distances. In the future, genome-wide molecular studies could be used to reconstruct range expansions to examine correspondence in phenotypic change with range expansion in the EDCS.

## (b) Relevance of punctuated evolution of learned song for speciation

The EDCS species complex bears hallmarks of speciation by sexual [3], or social [2] selection: species are strongly divergent for a signal used in social competition, and do not differ strongly in ecological respects [28]. Panhuis *et al.* [3] suggested that an additional signature of speciation by sexual selection is the evolution of variation in sexually selected traits among populations within species, with this variation generating partial pre-mating isolation. Our sampling of isolated sky island populations, especially within *C. moreaui* and *C. fuelleborni*, allows us to characterize within-species variation in territorial song. Across most song traits, variation across populations within species is minimal, including for many traits with strong differences across species, e.g. CV peak frequency (electronic supplementary material, figure SA5) and median pause duration (electronic supplementary material, figure A7). As such, between-species divergence cannot be predicted from within-species variation [49], suggesting that alternate evolutionary mechanisms contribute to divergence at different levels.

## 4. Conclusion

The effects of learning on evolutionary diversification processes are poorly explored for many organisms. Previous work has suggested that stabilizing selection on learned traits should be inadequate to prevent the divergence of genetic predispositions by drift, ultimately facilitating more rapid divergence in those genes underlying traits [12]. Our study shows that multiple song traits can exhibit stasis for prolonged periods, lasting hundreds of thousands of years or more. These results suggest that learned song in the focal taxa is subject to a combination of sufficiently strong stabilizing selection and sufficient exposure of the underlying genetic variation to prevent incremental change for long periods of time. An alternative, that there is insufficient genetic variation underlying these traits, is potentially plausible, but appears less likely given the evidence that genetic variation for learned song traits is present in other songbirds, and the long span of evolutionary time during which such variation could be generated.

## 5. Material and methods

### (a) Song analysis

We made sound recordings of EDCS from 2007 to 2011 in Kenya, Tanzania and Mozambique, using solid-state digital recorders (Marantz PMD models 660, 661 and 670) and shotgun microphones (Sennheiser ME-67). A small number of recordings were made using a parabolic dish with an omnidirectional microphone (Sennheiser ME-62). We complemented our field recordings with additional recordings from the Macaulay Library (http://macaulaylibrary.org) and the British Library of Natural Sounds (https://www.bl.uk/collection-guides/wildlife-and-environmental-sounds). The vocal repertoires of the focal taxa are complex, including a wide array of different signal types. Here, we measure the acoustic properties of male territorial songs delivered in bout form, in which consecutive songs are typically separated by a short duration (less than 15 s) of silence, or a series of short calls and pauses [50]. Sunbirds sing these songs from a perch in the vegetation, ranging in height from 2 to 30 m. These songs function in male–male territorial interactions [51]. Further, as in other passerine birds [15], these songs likely serve to attract mates. Singing can coincide with, or immediately precede, female wing-fluttering displays directed at singing males, which has been observed in *C. loveridgei* and *C. fuelleborni* (J.P.M. 2008 and 2009, personal observation).

Before analyses, recordings were standardized for frequency sampling at 44.1 kHz, and bandpass filtered at 2–10 kHz. More strict filtering, at 2.5–9 kHz, was then employed for recordings of *C. mediocris* and *C. usambaricus* to allow fine-scale structural analysis of sonograms, as our recordings of their songs generally had lower signal : noise ratios, and the lowest frequencies in their songs are greater than 2.5 kHz. Similarly, strict filtering could not be applied to *C. fuelleborni* or *C. moreaui* songs because their songs sometimes include peak frequencies below 2.5 kHz. Spot filtering was used to remove acoustic signals not emitted by the focal bird. We selected high-quality field recordings for analyses after sonogram visualization in Raven Pro 1.3 [52]. J.P.M. performed all sonogram analysis procedures in the program Luscinia [53]. Sonograms were produced in Luscinia with the following settings: maximum frequency: 10 kHz; frame length 5 ms; time step: 1 ms; spectrograph points: 221; spectrograph overlap: 80%; echo removal: 100%; echo range: 100; windowing function: Hann; and high-pass threshold: 2 kHz. Signals within sonograms were detected using Luscinia's automated signal detection. Results of automatic signal detection procedures were checked by eye and ear, with recordings slowed for playback to one-eighth speed. Automated signal detection errors were corrected using the *brush* tool. Measurements were made for each sonogram trace (hereafter 'elements'), separated by pauses from other elements.

From the set of measurements of each element, we calculated summary statistics at the song level. For each individual sunbird, we then calculated the mean values of a set of summary statistics across songs. We calculated the following summary statistics for each song, based on values for each element: median pause duration between elements (ms), coefficient of variation (CV) of pause duration, median peak frequency (Hz), CV peak frequency, maximum peak frequency (Hz), minimum peak frequency (Hz), range peak frequency (difference between maximum and minimum peak frequencies), number of elements, median frequency bandwidth (Hz), CV frequency bandwidth (Hz), median frequency change (Hz), CV per-element frequency change, song duration (ms) and median element duration (ms). The peak frequency is defined as the frequency window with the highest amplitude for a given portion of the sonogram. Because we extract peak frequencies from amplitude spectra for all our frequency measurements, our approach should not be subject to potential errors from manually selecting frequency windows [54]. To improve analyses with respect to assumptions of normality for non-phylogenetic analyses (linear mixed models and discriminant analyses), we took the natural log of those variables that were right-skewed. To generate estimates of song phenotypes at the level of the individual bird, we took the arithmetic mean of the values for each variable across songs. These procedures resulted in a dataset comprising song phenotype estimates for 142 individuals from measurements of 419 songs. A mean of $2.95 \pm .08$ s.e. songs were measured per individual.

To characterize sources of variation for song traits, we fit linear mixed models to the 14 song variables. These models each had an individual-level random effect nested within a population-level random effect. Populations represented distinct sky island forest patches. In performing these analyses, our first goal was to estimate fractions of *within-species* variance attributable to within-individual, among-individual and among-population variation. The within-individual variance is a component of the residual variance in our models. Our second goal was to identify which traits were most different among species. To accomplish this latter goal, we fit species as a fixed effect.

We performed discriminant analyses based on Gaussian finite mixture modelling on the 14 song traits measured for each individual, using the package Mclust [55] in R 4.1.0 [56]. We sought an optimal mixture model, by BIC, for discriminant analyses among the model types referred to in Mclust as (i) spherical, equal volume, (ii) spherical, unequal volume, (iii) diagonal, equal volume and shape, (iv) diagonal, equal volume, varying shape, (v) diagonal, equal volume, varying shape, and (vi) diagonal, varying volume and shape. We limited the number of mixture components to three per class. To visualize the results, we performed a dimension reduction using the function MclustDR, with default settings.

## (b) Molecular phylogenetics

We performed phylogenetic analyses using DNA sequence data for samples collected from the field (see the electronic supplementary material, appendix for details on sampling for molecular analyses and for further detail on phylogenetic methods, see electronic supplementary material, appendix 2 for specimen details). First, to investigate whether song phenotypes generally correspond to phylogenetic lineages across the species complex, we built a multi-locus phylogenetic tree from a concatenated alignment of DNA sequences for three mtDNA genes (ND2, ND3 and ATP6) and six nuclear autosomal introns (MB, CHDZ, 11836, 18142, TGFb2 and MUSK) for 256 in-group individuals and 12 outgroup species. The alignment had 5313 bp. We estimated our phylogenetic tree using a maximum-likelihood approach [57], and assessed support for nodes by bootstrapping. Secondly, we built population-level trees to investigate the history of song divergence over the focal species complex. We constructed these trees by calculating the mean population distances for 15 populations. For these trees, we used sequences of the mtDNA genes ND2 (440 bp) and ND3 (362 bp) for 134 individuals. Topological relationships between named species were the same in the mtDNA population trees as in the multi-locus phylogenetic tree with species as tips.

Third, we sought to estimate the age of the MRCA of the species complex. We performed a species-level phylogenetic analysis using a Bayesian coalescent-based method (BEAST, [58,59]) with DNA sequence data from two mtDNA genes (ND2 and ATP6, coded as a single locus) and four nuclear DNA sequences (ATP6, TGFb2, MB and CHDZ). This analysis included 16 species as tips, including the five named species in the focal species complex (*C. mediocris*, *C. usambaricus*, *C. loveridgei*, *C. moreaui* and *C. fuelleborni*), eight other sunbird species and three species of flowerpecker (Dicaeidae). We dated the MRCA of the focal species complex by implementing a normal prior distribution (mean = 18 My, s.d. = 2) on the node age of the MRCA of sunbirds and flowerpeckers, based on a recent dating analysis of a family-level phylogenetic tree of the Passeriformes [60]. We used a GTR-gamma substitution model and a relaxed lognormal molecular clock, with substitution rate prior distributions based on divergence rate estimates for the Hawaiian honeycreeper radiation (an oscine passerine radiation, like sunbirds) [61].

## (c) Phylogenetic comparative method approach

To investigate the tempo and mode of song divergence, we compared phylogenetic trait evolution models fit to population-level data. We fit models to each song trait individually. We built and fit phylogenetic trait evolution models representing (i) strongly bounded evolution (stasis or near-stasis) punctuated by pulses and (ii) gradual evolution (Brownian motion), by maximum likelihood. These models are described in the electronic supplementary material. Models including pulses were fit under the condition that there was a maximum of four pulses across the population tree. We compared support for fitted models using AICc values [62]. However, simulations indicated that

standard approaches to comparing AICc values were biased, so we developed AICc calibrations for each trait to compare model support (electronic supplementary material). Our approach may be understood as an approximate correction that takes into account phylogenetic correlations present in the data, as is necessary for BIC [63]. To characterize the uncertainty in model selection due to phylogenetic uncertainty, we built 10 bootstrap population trees, and fit both trait evolution models on each of the bootstrap phylogenies for each song trait.

For the traits where punctuated evolution models were a better fit than Brownian motion, we estimated the location of pulses on the population tree. Our method for fitting pulsed evolutionary models involves fitting the maximum-likelihood parameters for all potential pulse configurations on the tree, given a maximum of four pulses (corresponding to approx. $4.6 \times 10^4$ pulse configurations). To quantify the strength of the evidence for an evolutionary pulse on a given branch of the phylogenetic tree, we calculated the sum of the AICc weights of those pulse configurations that include a pulse on the given branch, and divided this by the sum of the AICc weights of all computed pulse configurations [64]. Our approach treats each $n$-pulse configuration as an independent model. We present maximum-likelihood parameters for the maximum-likelihood configuration (electronic supplementary material, tables S1 and S2).

**Ethics.** Field research permissions were furnished by: TAWIRI, COST-ECH and the Ministry of Natural Resources, Forestry and Beekeeping Division, Tanzania; National Museum Kenya, National Council for Science and Technology and Kenya Wildlife Service, Kenya; and the Universidade Eduardo Mondlane Natural History Museum of Maputo and Ministry of Agriculture, Mozambique. Animal handling procedures were approved by the IACUC committee at University of California, Berkeley, under protocols R317 and AUP-2016-04-8665.

**Data accessibility.** Novel DNA sequence data for molecular phylogenetics have been deposited in GenBank for the following genes, with corresponding accession numbers: BRM—MT235540–MT235555; CHDZ and MB—MT327145–MT327175; MUSK—MT319734–MT319749; ATP6—MN897893–MN897917; ND2 and ND3—MT513326–MT513717. Song data and code for performing analyses are available at https://github.com/hlebzh/birdsongs.

**Authors' contributions.** J.P.M.: conceptualization, data curation, formal analysis, funding acquisition, investigation, methodology, project administration, supervision, visualization, writing—original draft, writing—review and editing; G.Z.: conceptualization, formal analysis, investigation, methodology, software, visualization, writing—review and editing; C.W.: investigation, project administration, resources, writing—review and editing; N.N.: data curation, investigation, writing—review and editing; J.V.P.: investigation, writing—review and editing; E.M.: investigation, resources; M.M.: investigation, resources; S.K.: investigation; L.C.: methodology, resources, supervision; J.G.B.: methodology, resources, supervision, writing—review and editing; R.C.K.B.: conceptualization, funding acquisition, investigation, methodology, project administration, resources, supervision, writing—review and editing.

All authors gave final approval for publication and agreed to be held accountable for the work performed therein.

**Competing interests.** We declare we have no competing interests.

**Funding.** The work was supported by the National Geographic Society; Waitt Foundation; Mohamed bin Zayed Species Conservation Fund; a National Science Foundation (NSF) Doctoral Dissertation Improvement Grant (award no. 1011702); a Beim Summer Research Award (UC Berkeley Department of Integrative Biology), Louise Kellogg, Albert Preston Hendrickson and Charles Koford grants from the Museum of Vertebrate Zoology; a UC Berkeley Center for African Studies Andrew and Mary Thompson Rocca Scholarship; American Ornithologists' Union (now American Ornithological Society); the Explorers Club; and NSF grant no. DBI-1458034. G.Z. and L.C. were supported by the Leverhulme trust grant no. RPG-2017-249. G.Z. was supported by NSF grant no. DMS-1056471. L.C. was supported by the Royal Society of Edinburgh and the Scottish Government Personal Research Fellowship.

**Acknowledgements.** We thank Jessica Hughes, Violet Kimzey, Somin Lim, Kyle Marsh, Dalila Sequeira, Emilia Wakamatsu, Cynthia Wang and Addien Wray for research assistance. Kathleen Rudolph assisted by designing and illustrating figure 1. Ruth King contributed advice on statistical issues. David Moyer, Norbert Cordeiro, Liz Baker and Neil Baker provided advice and material support for fieldwork. Caffe Luce in Tucson, Arizona, provided work space.

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
