## [Peer Review File · Proceedings of the Royal Society B: Biological Sciences]

Review History

RSPB-2020-1559.R0 (Original submission)

Review form: Reviewer 1

Recommendation

Major revision is needed (please make suggestions in comments)

Scientific importance: Is the manuscript an original and important contribution to its field?

Good

General interest: Is the paper of sufficient general interest?

Acceptable

Quality of the paper: Is the overall quality of the paper suitable?

Marginal

Is the length of the paper justified?

No

Should the paper be seen by a specialist statistical reviewer?

Yes

Do you have any concerns about statistical analyses in this paper? If so, please specify them explicitly in your report.

No

It is a condition of publication that authors make their supporting data, code and materials available - either as supplementary material or hosted in an external repository. Please rate, if applicable, the supporting data on the following criteria.

Is it accessible?

Yes

Is it clear?

Yes

Is it adequate?

Yes

Do you have any ethical concerns with this paper?

No

Comments to the Author

Highlights

Past work has focused on the role of learned song as a pre-mating isolating mechanism and fewer studies have asked about the tempo and mode of song evolution. Those that have focused on gradual evolution. The authors use a fascinating songbird system to ask whether gradual or punctuated evolution best describes the evolution of song traits. Overall, this is an interesting idea and the methods appear sound to me (a few comments below). However, I think that the authors need to scale back the introduction and the discussion to align with the scope of the methods and results. I also think there is a strong emphasis on discussing those song traits which showed punctuated evolution, when these were actually in the minority. This is not to say that these traits aren't interesting – evidence of punctuated evolution is quite interesting – but I would make sure to discuss the evidence for gradual evolution as well.

Major comments

Introduction is very thorough, which is great, but reads more like a review (which I think is actually a much needed review, and one the authors should consider separately perhaps?). I suggest paring the introduction back to focus on setting up the specific points that are tested in this manuscript.

I think the final paragraph could be better focused with a clear hypo-deductive framework. There are statements about methods (multivariate and univariate approaches) but it isn't clear how these approaches link to a question set up earlier in the intro. Then the final paragraph turns to a point about gradual versus punctuated evolution. This makes it sound like this is the main and only question tested in this study. Although there is a nice set up for why one might expect gradual evolution, the set up for punctuated evolution is not as clear. This is particularly surprising given the general depth of the introduction overall. When I read the final paragraph, I was left wondering whether the study was going to ask any of the other questions brought up earlier in the introduction. I was also left wondering how the referenced methods (multivariate and univariate) specifically mapped onto testing gradual versus punctuated evolution.

In your discussion of the sources of stabilizing selection, you mention male and female receivers. The way in which this is discussed makes it sound like selection is occurring in the contexts of mate choice and male-male competition. Another source of stabilizing selection on these cultural

traits is the learning process itself. The mode of song learning could be strongly stabilizing if there is conformity bias in the learning process (see papers by Rob Lachlan on this point).

There is an interesting suggestion that range shifts could drive punctuated evolution, but, unless I'm mistaken, there is on average one shift in song traits whereas I would imagine there would be more range shifts over time. How might the placement of shifts in song evolution tell us what might underly these shifts?

One component missing is a discussion of those song traits that exhibit gradual evolution. What of those? Why might some song traits show punctuated but most show gradual evolution? How might that comparison give insight into what might be driving punctuated evolution?

Minor comments

Line 194 – the text here suggests that Fig 2 will illustrate the degree to which pulses are co-localized, but it appears that only song duration is plotted in figure 2. What is missing?

Line 217-291 This sentence is awkwardly phrased and a bit hard to follow as such.

Line 253 – 256 This is another sentence that is quite densely worded and thus hard to follow. Perhaps unpack into multiple, simpler sentences?

Line 266 – At this point, I realized that you hadn't actually mentioned the song traits exhibiting punctuated evolution – are these temporal or spectral traits or both? Would help to know as you start to discuss potential mechanisms.

Line 283 – 286 – I see your reasoning in here but all of this is conjecture as you don't include any analyses that actually test how morphological traits are evolving or for correlations among morphological and song traits. I think your hypothesis is highly plausible, particularly as morphological traits like body size are much less of a constraint on frequency characters in songbirds (which control frequency using syringeal muscles) than in suboscines, for example, which are more likely to show correlated evolution of bod size and peak frequency, but I think you have to be careful not to conjecture too far, particularly without analyses to support these points.

Line 302 – The traits showing pulsed evolution are mentioned here – I would make sure to state these clearly in the results section.

Line 303 – I agree that studies have found evidence of underlying genetic variation for fine-scale temporal and duration characters. I am not convinced by frequency, and which frequency trait do you mean here? All frequency traits?

Line 325 – Can you be more specific by what you mean here regarding 'discontinuities in evolutionary processes'?

Line 331 – I would be careful with this first sentence as there is substantial work on this idea, beginning with the work of Cavalli-Sforza and Feldman in humans, that provides quite a bit of insight into how learning shapes evolutionary processes. I agree that less is known in birds.

Methods

Overall, the methods look spot on to me. *Luscinia* is a great program to use, especially for field recordings that have low signal to noise ratios. I would make sure to make a statement about how you dealt with common issues of taking frequency measurements (e.g., signal amplitude and signal distortion effects on frequency measures – see (1))

References

1. S. A. Zollinger, J. Podos, E. Nemeth, F. Goller, H. Brumm, On the relationship between, and measurement of, amplitude and frequency in birdsong. *Animal Behaviour* 84, e1-e9 (2012).

Review form: Reviewer 2

Recommendation

Major revision is needed (please make suggestions in comments)

Scientific importance: Is the manuscript an original and important contribution to its field?

Marginal

General interest: Is the paper of sufficient general interest?

Excellent

Quality of the paper: Is the overall quality of the paper suitable?

Marginal

Is the length of the paper justified?

Yes

Should the paper be seen by a specialist statistical reviewer?

Yes

Do you have any concerns about statistical analyses in this paper? If so, please specify them explicitly in your report.

No

It is a condition of publication that authors make their supporting data, code and materials available - either as supplementary material or hosted in an external repository. Please rate, if applicable, the supporting data on the following criteria.

Is it accessible?

N/A

Is it clear?

N/A

Is it adequate?

N/A

Do you have any ethical concerns with this paper?

No

Comments to the Author

These comments are in order of line number, not importance. Some important conceptual comments are at the end. I have no comments on the phylogenetic methods which are outside of my expertise.

55- "long been thought" might cite Mayr 1963 book on Animal Species and Evolution

58 - Mayr there too, on reproductive isolation.

68 "learned signals are potentially subject to different evolutionary pressures" - Need to make more precise statements than this since "evolutionary pressures" is meaningless by itself. What you could mean and should say is that learned signals (learned phenotypes) are subject to non-genetic variation. As written a reader might think that selection operates differently on them (people like to refer, vaguely, to "selection pressure"), which would not be true. Selection sees phenotypes, including learned ones, without any regard to underlying mechanism of variation,

whether learning or genetic.

68 "...may exhibit different evolutionary trajectories" - similarly vague. Say how you think they might be distinctive! If you don't have a specific meaning in mind then leave this out - giving a reference does not clarify the sentence (the reader should not have to read another paper to find out).

70-71 "...cultural novelties appear frequently," This would make sense if it said "cultural (learned) novelties may be repeated frequently and therefore have their potentially complex properties be subject to selection frequently compared to those, eg. mutationally-mediated traits, that require positive selection to spread and have their complex features subject to selection." If you mean something else please state it precisely.

79 - "specific conditions" such as what? Again, the thought starts off well but becomes vague in the end. The reader has to keep guessing at what your point is supposed to be.

87 - species don't learn. Do you mean that individuals of a particular species have species-specific limits on what they learn?

132 ff - can you say "pattern" or even "phenotypic pattern" rather than "mode?" I realize that tempo and mode were successfully used by G. G. Simpson, e.g. in a book title, but these days "mode" could mean drift vs selection. Aren't you going to be looking for patterns and even recurrent patterns of phenotypic change? Again - please give the reader a concrete image of what you are trying to say.

136 - maybe this question has not been put into the dichotomy you pose here, but bird song and plumage diversification can be seen in terms of continuous change under sexual selection only leading to diversification when speciation (population isolation, as on an island or a mountain top) is possible, and the dilution of local changes by mobility among sites is curtailed. That is, gradualism can lead to "punctuation" depending on geographic conditions. The "mode" of evolution has not changed, only the setting - you cannot have diversification without isolation of "gene pools" (breeding populations).

Discussion

241 "here was little reason to expect learned songs to be restricted to peaks, because of their high variability" I agree, but for a different reason: "adaptive peaks" imply a zone of the environment where a particular phenotype is adaptive under natural selection. Selection on signals - social or sexual selection - does not follow that model because it concerns success in social competition, not with respect to some aspect of the environment other than the social environment (composed of conspecifics with which individuals interact locally).

An hypothesis to explain a punctuated pattern of signal evolution that is not discussed here (it is not covered by range expansion and founder effect although they might play a role) is that stasis persists in panmictic populations and a burst of evolutionary change occurs when a population becomes isolated from others, e.g. colonizes an island or a mountain top and becomes isolated there. I could not tell from this analysis what the geographic histories of the various lineages was: were the closely related sub-populations (newly formed species) within clades isolates, and the species showing stasis large or old populations without barriers to inbreeding within them for long periods of time? And were the divergent populations geographic isolates, perhaps newer than the populations showing stasis? That would support the general picture of how diversification and speciation looks under sexual or social selection on traits like territorial signals. The adaptive peak analogy does not seem appropriate to me for reasons just explained.

BUT there is an opportunity to use these data to show how a punctuated pattern might occur under different conditions - panmixis (stasis) vs isolation (rapid divergence) - and much

literature on island birds etc with a similar pattern might be cited to support it.

Decision letter (RSPB-2020-1559.R0)

21-Aug-2020

Dear Dr McEntee:

I am writing to inform you that your manuscript RSPB-2020-1559 entitled "Punctuated evolution in the learned songs of African sunbirds" has, in its current form, been rejected for publication in Proceedings B.

This action has been taken on the advice of referees, who have recommended that substantial revisions are necessary. With this in mind we would be happy to consider a resubmission, provided the comments of the referees are fully addressed. However please note that this is not a provisional acceptance.

Sincerely,
Dr Sasha Dall
<mailto:proceedingsb@royalsociety.org>

Associate Editor
Board Member: 1
Comments to Author:

I agree with the reviewers that the results of this study are intriguing and the model system is great for addressing evolutionary patterns of learned traits. Both reviewers have excellent insights, all of which should be addressed. In particular, both are looking for more in-depth discussions of the role of geographic range changes and the role of learning in this process. Authors also should modify the introduction and discussion as suggested, including

opportunities for clarification and a more balanced discussion of evidence for both gradual and punctuated evolutionary patterns.

Reviewer(s)' Comments to Author:

Referee: 1

Comments to the Author(s)

Highlights

Past work has focused on the role of learned song as a pre-mating isolating mechanism and fewer studies have asked about the tempo and mode of song evolution. Those that have focused on gradual evolution. The authors use a fascinating songbird system to ask whether gradual or punctuated evolution best describes the evolution of song traits. Overall, this is an interesting idea and the methods appear sound to me (a few comments below). However, I think that the authors need to scale back the introduction and the discussion to align with the scope of the methods and results. I also think there is a strong emphasis on discussing those song traits which showed punctuated evolution, when these were actually in the minority. This is not to say that these traits aren't interesting – evidence of punctuated evolution is quite interesting – but I would make sure to discuss the evidence for gradual evolution as well.

Major comments

Introduction is very thorough, which is great, but reads more like a review (which I think is actually a much needed review, and one the authors should consider separately perhaps?). I suggest paring the introduction back to focus on setting up the specific points that are tested in this manuscript.

I think the final paragraph could be better focused with a clear hypo-deductive framework.

There are statements about methods (multivariate and univariate approaches) but it isn't clear how these approaches link to a question set up earlier in the intro. Then the final paragraph turns to a point about gradual versus punctuated evolution. This makes it sound like this is the main and only question tested in this study. Although there is a nice set up for why one might expect gradual evolution, the set up for punctuated evolution is not as clear. This is particularly surprising given the general depth of the introduction overall. When I read the final paragraph, I was left wondering whether the study was going to ask any of the other questions brought up earlier in the introduction. I was also left wondering how the referenced methods (multivariate and univariate) specifically mapped onto testing gradual versus punctuated evolution.

In your discussion of the sources of stabilizing selection, you mention male and female receivers. The way in which this is discussed makes it sound like selection is occurring in the contexts of mate choice and male-male competition. Another source of stabilizing selection on these cultural traits is the learning process itself. The mode of song learning could be strongly stabilizing if there is conformity bias in the learning process (see papers by Rob Lachlan on this point).

There is an interesting suggestion that range shifts could drive punctuated evolution, but, unless I'm mistaken, there is on average one shift in song traits whereas I would imagine there would be more range shifts over time. How might the placement of shifts in song evolution tell us what might underly these shifts?

One component missing is a discussion of those song traits that exhibit gradual evolution. What of those? Why might some song traits show punctuated but most show gradual evolution? How might that comparison give insight into what might be driving punctuated evolution?

Minor comments

Line 194 – the text here suggests that Fig 2 will illustrate the degree to which pulses are co-localized, but it appears that only song duration is plotted in figure 2. What is missing?

Line 217-291 This sentence is awkwardly phrased and a bit hard to follow as such.

Line 253 – 256 This is another sentence that is quite densely worded and thus hard to follow. Perhaps unpack into multiple, simpler sentences?

Line 266 – At this point, I realized that you hadn't actually mentioned the song traits exhibiting punctuated evolution – are these temporal or spectral traits or both? Would help to know as you start to discuss potential mechanisms.

Line 283 – 286 – I see your reasoning in here but all of this is conjecture as you don't include any analyses that actually test how morphological traits are evolving or for correlations among morphological and song traits. I think your hypothesis is highly plausible, particularly as morphological traits like body size are much less of a constraint on frequency characters in songbirds (which control frequency using syringeal muscles) than in suboscines, for example, which are more likely to show correlated evolution of bod size and peak frequency, but I think you have to be careful not to conjecture too far, particularly without analyses to support these points.

Line 302 – The traits showing pulsed evolution are mentioned here – I would make sure to state these clearly in the results section.

Line 303 – I agree that studies have found evidence of underlying genetic variation for fine-scale temporal and duration characters. I am not convinced by frequency, and which frequency trait do you mean here? All frequency traits?

Line 325 – Can you be more specific by what you mean here regarding 'discontinuities in evolutionary processes'?

Line 331 – I would be careful with this first sentence as there is substantial work on this idea, beginning with the work of Cavalli-Sforza and Feldman in humans, that provides quite a bit of insight into how learning shapes evolutionary processes. I agree that less is known in birds.

Methods

Overall, the methods look spot on to me. *Luscinia* is a great program to use, especially for field recordings that have low signal to noise ratios. I would make sure to make a statement about how you dealt with common issues of taking frequency measurements (e.g., signal amplitude and signal distortion effects on frequency measures – see (1))

References

1. S. A. Zollinger, J. Podos, E. Nemeth, F. Goller, H. Brumm, On the relationship between, and measurement of, amplitude and frequency in birdsong. *Animal Behaviour* 84, e1-e9 (2012).

Referee: 2

Comments to the Author(s)

These comments are in order of line number, not importance. Some important coceptural comments are at the end. I have no comments on the phylogenetic methods which are outside of my expertise.

55- "long been thought" might cite Mayr 1963 book on *Animal Species and Evolution*

58 – Mayr there too, on reproductive isolation.

68 "learned signals are potentially subject to different evolutionary pressures" - Need to make more precise statements than this since "evolutionary pressures" is meaningless by itself. What you could mean and should say is that learned signals (learned phenotypes) are subject to non-genetic variation. As written a reader might think that selection operates differently on them (people like to refer, vaguely, to "selection pressure"), which would not be true. Selection sees phenotypes, including learned ones, without any regard to underlying mechanism of variation, whether learning or genetic.

68 "...may exhibit different evolutionary trajectories" - similarly vague. Say how you think they might be distinctive! If you don't have a specific meaning in mind then leave this out – giving a

reference does not clarify the sentence (the reader should not have to read another paper to find out).

70-71 "...cultural novelties appear frequently," This would make sense if it said "cultural (learned) novelties may be repeated frequently and therefore have their potentially complex properties be subject to selection frequently compared to those, eg. mutationally-mediated traits, that require positive selection to spread and have their complex features subject to selection." If you mean something else please state it precisely.

79 - "specific conditions" such as what? Again, the thought starts off well but becomes vague in the end. The reader has to keep guessing at what your point is supposed to be.

87 - species don't learn. Do you mean that individuals of a particular species have species-specific limits on what they learn?

132 ff - can you say "pattern" or even "phenotypic pattern" rather than "mode?" I realize that tempo and mode were successfully used by G. G. Simpson, e.g. in a book title, but these days "mode" could mean drift vs selection. Aren't you going to be looking for patterns and even recurrent patterns of phenotypic change? Again - please give the reader a concrete image of what you are trying to say.

136 - maybe this question has not been put into the dichotomy you pose here, but bird song and plumage diversification can be seen in terms of continuous change under sexual selection only leading to diversification when speciation (population isolation, as on an island or a mountain top) is possible, and the dilution of local changes by mobility among sites is curtailed. That is, gradualism can lead to "punctuation" depending on geographic conditions. The "mode" of evolution has not changed, only the setting - you cannot have diversification without isolation of "gene pools" (breeding populations).

Discussion

241 "here was little reason to expect learned songs to be restricted to peaks, because of their high variability" I agree, but for a different reason: "adaptive peaks" imply a zone of the environment where a particular phenotype is adaptive under natural selection. Selection on signals - social or sexual selection - does not follow that model because it concerns success in social competition, not with respect to some aspect of the environment other than the social environment (composed of conspecifics with which individuals interact locally).

An hypothesis to explain a punctuated pattern of signal evolution that is not discussed here (it is not covered by range expansion and founder effect although they might play a role) is that stasis persists in panmictic populations and a burst of evolutionary change occurs when a population becomes isolated from others, e.g. colonizes an island or a mountain top and becomes isolated there. I could not tell from this analysis what the geographic histories of the various lineages was: were the closely related sub-populations (newly formed species) within clades isolates, and the species showing stasis large or old populations without barriers to inbreeding within them for long periods of time? And were the divergent populations geographic isolates, perhaps newer than the populations showing stasis? That would support the general picture of how diversification and speciation looks under sexual or social selection on traits like territorial signals. The adaptive peak analogy does not seem appropriate to me for reasons just explained.

BUT there is an opportunity to use these data to show how a punctuated pattern might occur under different conditions - panmixis (stasis) vs isolation (rapid divergence) - and much literature on island birds etc with a similar pattern might be cited to support it.

Author's Response to Decision Letter for (RSPB-2020-1559.R0)

See Appendix A.

RSPB-2021-0498.R0

Review form: Reviewer 2

Recommendation

Accept with minor revision (please list in comments)

Scientific importance: Is the manuscript an original and important contribution to its field?

Excellent

General interest: Is the paper of sufficient general interest?

Excellent

Quality of the paper: Is the overall quality of the paper suitable?

Excellent

Is the length of the paper justified?

Yes

Should the paper be seen by a specialist statistical reviewer?

Yes

Do you have any concerns about statistical analyses in this paper? If so, please specify them explicitly in your report.

No

It is a condition of publication that authors make their supporting data, code and materials available - either as supplementary material or hosted in an external repository. Please rate, if applicable, the supporting data on the following criteria.

Is it accessible?

Yes

Is it clear?

N/A

Is it adequate?

N/A

Do you have any ethical concerns with this paper?

No

Comments to the Author

The initial, background section nicely points out why the dynamics of change in learned song are likely to differ from, and relate to, genetic change, under selection. The paper describes important and original insights, capitalizing on data on territorial/mate-attracting song in a group of birds that can be used to investigate the relationship between learned and evolutionary-genetic changes in song. Looking at these questions with an overarching phylogenetic view, while

relating it to the more usual micro-evolutionary perspective, and using a particular example (sunbirds) is mind-expanding for anyone interested in the evolution of social signals, especially those influenced by learning. Given the complexity of the questions, this paper manages to clearly work through the many conceptual complications to an admirable degree.

The one very small clarifying suggestion I have is that it should be pointed out as soon as the word "territorial" is used that the calls that are the focus of this study function as both territorial and mate-attraction signals. Many of the points discussed, such as relevance of song to reproductive isolation and speciation, usually refer to courtship song and female choice, which has different dynamics than do threat and exclusively territorial (e.g., aggressive) signals. [Due to the genetic correlation between signals and choice in courtship signaling the evolution of courtship songs are expected to change more rapidly and attain greater complexity than territorial signals, which are threat signals whose dynamics include imposition of "truth" by ability to back up threats with real strength.] Given the importance of this distinction the place to point this out is on line 106: change that sentence to state that the paper focuses on "learned territorial/mate attracting songs" of sunbirds.

Small suggestions:

Ln 37 - "dynamics" might be better than "trajectories"

40 - longer should be "long"

Ln 116 say in a parenthesis what "evolutionary mode" means, e.g. (punctuation, stasis, gradual change)

122 change "of" to for, or regarding

131 - can you change "multifarious" to "not uniform?" Or just variable or "varies in its nature," which better relates to the next sentence.

Review form: Reviewer 3

Recommendation

Reject - article is scientifically unsound

Scientific importance: Is the manuscript an original and important contribution to its field?

Marginal

General interest: Is the paper of sufficient general interest?

Acceptable

Quality of the paper: Is the overall quality of the paper suitable?

Poor

Is the length of the paper justified?

No

Should the paper be seen by a specialist statistical reviewer?

No

Do you have any concerns about statistical analyses in this paper? If so, please specify them explicitly in your report.

No

It is a condition of publication that authors make their supporting data, code and materials available - either as supplementary material or hosted in an external repository. Please rate, if applicable, the supporting data on the following criteria.

Is it accessible?

Yes

Is it clear?

Yes

Is it adequate?

No

Do you have any ethical concerns with this paper?

No

Comments to the Author

The African sunbirds appear to be an excellent system to study song evolution, given the six (or so) species and separated populations within each species, along with the presence of some contact zones. The authors of this manuscript have collected an impressive number of song recordings (419 songs from 142 individuals), setting them up for a reasonably comprehensive survey of song variation within and among the 6 species. This manuscript uses a mapping of song variation onto a molecular phylogeny to conclude that a model with bursts of evolution fits the song variation better than a model of random change at a constant expected rate (Brownian motion). The authors' interpretation of this result is that songs show a "punctuated evolution" pattern of song variation, and that songs have experienced long periods ("> 10⁶ years") of near stasis.

Unfortunately, this conclusion is not convincing given the analyses and data presented. The main signal for the "burst" of evolution comes from the comparison of the fuelleborni species compared to the others. For instance, Fig. 2 shows that the 5 populations of fuelleborni have similarly long songs compared to most other species in the phylogeny, and so a comparison of the two models indicates support for the burst model over the Brownian motion model. There is however an alternative explanation that has not been sufficiently considered by the manuscript: Movement of individuals between the different fuelleborni populations could have spread a similar song type between those populations. This could happen as a result either of genetic exchange or purely memetic (transmission of traits through learning) exchange, and could be a result of selection (a new song type is advantageous) or mutation and drift alone. It is not clear why the manuscript concludes from the similarity of song in these populations that songs have been in stasis for more than a million years. Much of the reasoning appears to be based on the molecular phylogeny suggesting the populations started diverging that long ago, but the genetic data were analyzed with a method that can only present the data in a bifurcating phylogeny, rather than methods that have the capability to indicate genetic or memetic exchange between populations.

Another reason that stasis is not very convincing is that there is in fact much within-species variation in song. This is mentioned in the text, and is visible in Figure A1. The bivariate plots in that figure show much variation within species, and much overlap between clouds of points for different species. In particular fuelleborni shows much variation, and overlap or contact in acoustic space with other species.

The analysis considers two models of song variation, and it is worth considering whether there are others that may be more consistent with the data and the evolutionary process. A model that was not considered is one in which there is Brownian motion but the rate of that Brownian motion can itself change gradually through the phylogeny. This is likely more consistent with what is known about song evolution. If that is truly how songs are evolving, then the constant-

rate Brownian motion model would not be well supported, and the alternative in this manuscript of a burst model might then appear to be favored in comparison. In other words, when only comparing two models, one might look better than the other but they might both be far from the truth.

Although the manuscript is about song evolution, the manuscript is actually not very clear about the pattern of variation in song. It is difficult to see details and compare songs in the Fig. 2C, and showing just a single song for each species seems insufficient given the apparent variation in songs as indicated in Fig. A1. It seems that examining and somehow presenting a summary of song variation within individuals, between individuals within a population, and among populations is important for the reader to understand the overall context of song variation, and only then can hypotheses regarding the evolutionary process be effectively addressed.

A few other recommendations:

It would be good if the different figures were more integrated in terms of terminology, use of color and symbols, etc. For example, it is presently unclear whether the mediocris N vs. S in Fig. 1 corresponds to MtKenya vs. Mbulu in Fig. 2. The colors used for taxa in Fig. A1 are different than those used in Fig. 1.

There are a variety of phylogenies used in the main paper and the appendix and supplement, and the captions often don't give many details about the source of the data and the method that built the phylogeny. This makes it difficult for the reader to understand the different appearance of the different phylogenies (e.g. compare Fig. A2, where variation among most pops of feulleborni seems very low; vs. Fig. 2, where it looks much deeper).

The paper mostly presents univariate analyses of the different song variables – it says that a multivariate clustering approach generated distinct clusters, but there doesn't seem to be a figure showing those clusters in a way that the reader can evaluate the support for them (e.g., how distinct were they, how much overlap). It would be good if some sort of figure such as a PCA or similar were shown.

Overall, the authors appear to have a valuable dataset on song variation in this interesting group of birds. A paper that presents the patterns of variation more clearly to the reader and that considers other models than just the two here would be of much value. In particular, it would be good if the authors considered (1) the possibility of the present song variants having spread among populations within a species rather recently, rather than evolving strictly through a bifurcating phylogeny, and (2) the possibility that neither the constant-rate Brownian motion nor the burst model being the best models of real song evolution.

Review form: Reviewer 4

Recommendation

Major revision is needed (please make suggestions in comments)

Scientific importance: Is the manuscript an original and important contribution to its field?

Good

General interest: Is the paper of sufficient general interest?

Excellent

Quality of the paper: Is the overall quality of the paper suitable?

Acceptable

Is the length of the paper justified?

Yes

Should the paper be seen by a specialist statistical reviewer?

Yes

Do you have any concerns about statistical analyses in this paper? If so, please specify them explicitly in your report.

Yes

It is a condition of publication that authors make their supporting data, code and materials available - either as supplementary material or hosted in an external repository. Please rate, if applicable, the supporting data on the following criteria.

Is it accessible?

Yes

Is it clear?

Yes

Is it adequate?

Yes

Do you have any ethical concerns with this paper?

No

Comments to the Author

General comments:

This study introduces a novel technique for measuring pulsed phenotypic evolution, and applies it to a fascinating empirical system: the evolution of culturally evolving song in an 'land island' species complex. The paper is therefore ambitious, and of potentially broad impact on two fronts.

(1) As someone with only shallow understanding of comparative phylogenetic methods, I found it difficult to assess the new technique for measuring pulsed evolution. It seems to me that the new method could easily have been presented as a standalone paper, and relegating it to a supplemental materials has perhaps meant that less effort has gone into making it easy to follow for interested readers with a limited expertise, like me. I would have appreciated, for example, an explanation of how this approach compares to related (I think) methods examining different evolutionary rates within a phylogenetic tree by O'Meara et al, Revell etc. I also think that more could have been done to explain the figures, especially S1, S2, S6 and S7 - even as far as what high values of var_d and var_p imply. Finally, since this is the introduction of a brand new method in a field where previous studies have been controversial, I think that the paper would have benefitted from efforts to examine its performance with simulated data. I realise that you have simulated data to correct for AICc, but would it make sense to go one step further and demonstrate that this correction is itself reliable, by applying the whole analysis pipeline to simulated data? If given phenotypic data simulated under Brownian evolution, what is the probability that the method (in toto) would erroneously detect pulsed evolution?

(2) The paper examines phenotypic evolution of song over a molecular phylogenetic tree of a species complex. I did not find any discussion of potential gene flow between populations - even though this is a closely related group of populations. I think this is potentially problematic for the statistical inference of pulsed phenotypic evolution. Gene flow would, of course, tend to homogenise populations in contact, and give an appearance of comparatively ancient divergence between them and populations not in contact. In other words: gene flow could generate an

illusion where a recent selective sweep across populations would appear as rapid evolution before the MRCA. In fact, one example of such gene flow is explicitly mentioned in the Discussion (Lines 258-260)

Tests for gene flow should be easy to carry out, and my concern easily dealt with. In this particular case, the situation is potentially more tricky because song might spread between population boundaries even when genes don't - through cultural evolution.

An alternative explanation for the pattern of phenotypic divergence that is described is that song is regularly being shared across population boundaries, and that geographic distance is therefore a better predictor of song similarity than phylogenetic distance. It would be great if the authors could test and refute that alternative. I realise that there is some possible evidence that songs may not pass across one population boundary, but can that be applied to the critical boundaries *within* the fuelleborni clade, where most of the pulsed evolution seems to occur?

(3) At some points, I felt that the manuscript conflated the acoustic measurements that were made with the cultural units that were transmitted from one individual to another. Individual song-types or syllables are learned by birds; in songs like these, they are very high-dimensional trajectories of how frequency changes over time. Measurements such as "mean peak frequency" obviously only capture a small proportion of that variation: I'm sure there are many different song-types in these populations that share the same mean peak frequency. Many of the 14 acoustic features that were measured are inter-correlated, and together, I would bet that they only capture a small proportion of the variation in song structure.

This impinges on the interpretations in the manuscript in several ways. First of all, broad-scale acoustic measurements, like those used here, are likely to reflect general constraints acting on learning (caused by vocal tract morphology, environmental selection, or perceptual predispositions). I would argue that it's the norm in other species that such measurements are stable over large periods of time. For example, studies of several species have explicitly demonstrated that general song features are constant over continent-wide scale populations of millions of individuals (e.g. black-capped chickadees, chaffinches, chiffchaffs, swamp sparrows, just to name a few). This geographic consistency implies considerable stasis over time too.

So in the Introduction and the Discussion (e.g. L246-249), it feels to me that the manuscript constructs straw men. There should have been clear a priori expectations that song features such as duration and peak frequency show stasis over long periods of time *because* these likely reflect stable constraints on learning, rather than cultural evolution itself, and *because* evidence from other species shows considerable stability over large distances.

(4) Learned bird songs are typically thought to develop through a combination of genetically transmitted constraints (in terms of what is perceived as conspecific, and what can be produced with the vocal organs) and cultural transmission. It is not immediately clear what these results suggest in this regard. It could be the case that the pulses of change represent evolution in underlying constraints. The manuscript is somewhat unclear about this - it acknowledges that predispositions and constraints exist, but focuses on the learned aspect of song, but then brings up predispositions again in the conclusions. I would personally prefer it if the manuscript was a little more explicit.

(5) I think the manuscript neglects literature on learned bird song and speciation. This applies both to modelling studies (by Servedio and colleagues), but also to other studies of island diversification and bird song.

Specific Comments

Line 72 - nb - I don't think that the reference cited here reports original empirical findings. I am not aware of any strong evidence that novel songs are at a selective advantage (although this has frequently been proposed)

Line 82 - quite a lot of empirical data suggests an alternative pattern, however: that the "acoustic space" permitted by genetic predispositions becomes saturated within even moderately sized populations, such that little differentiation is observed between isolated populations. It feels somewhat as if you are setting up a straw man here.

Lines 84 - but explicit models of the role of song learning in speciation are not cited. This includes Lachlan & Servedio (which is cited only in the Conclusions), but also other modelling studies by Servedio.

Lines 92-98 - It seems to me that - as fantastic a study as it is - there is rather too much space taken with discussing the white-throated sparrow study. There are plenty of other studies showing patterns relevant to this one, especially long-term stasis (e.g. Pipek et al in yellowhammers, Lachlan et al. in swamp sparrows).

Lines 87-95 - These two examples seem to be confusing two different levels of variation. In greenish warblers, broad scale acoustic features were measured - such as minimum frequency. These measures might be expected to reflect broad constraints on song variation, but not to identify individual song-types.

In contrast, the white-throated sparrow study looks at a particular song-type - albeit with very constrained species levels of variation.

Another key example of pulsed variation is humpback whale song (e.g. Garland et al 2011, Curr. Biol.)

Lines 171-179. Your statistical analysis did not seem to take into account any correction for multiple comparisons. However, you are comparing 14 song traits (some which are likely to be inter-correlated), without any a priori hypotheses for which should show pulsed evolution. Therefore, it would seem appropriate to me that your statistical discovery of pulses should take into account the risk of an inflated type II error. If you were to simulate 14 phenotypic traits under Brownian phenotypic evolution, what would be the probability of finding 4 show clear signs of pulsed evolution?

Line 185 - It would be useful to provide a composite tree in the main paper showing the placement of pulses across different traits.

Line 205. Studies of stasis in learned bird song have been carried out: Pipek et al on yellowhammers; Lachlan et al. 2018 on swamp sparrows, among them.

Line 312. I think this is clumsily phrased: there is no empirical whatsoever for genetic assimilation of song traits. The sources mentioned write about genetic assimilation much more generally.

Supplemental Information

At the foot of P6 there must be a typo / missing words in the sentence beginning "Informally..." Please check throughout for typos - in some places I found it hard to follow I think because of similar typos.

Table S2 - I found this confusing. Delta A is used both to refer to the four CI columns, but then there is a separate Delta A column, which is not explained.

Fig. S5 - the axis of these figures is not described, nor the units.

Fig. S8 - the scale bar is not explained. The figure legend should carefully explain what the thick bar means, and what it was based on. After reading the supplemental materials carefully, I can work most of it out, but the figure legend should make it more explicit.

Decision letter (RSPB-2021-0498.R0)

16-Apr-2021

I am writing to inform you that this version of your manuscript RSPB-2021-0498 entitled "Punctuated evolution in the learned songs of African sunbirds" has, in its current form, been rejected for publication in Proceedings B.

This action has been taken on the advice of referees, who have recommended that substantial revisions are necessary. With this in mind we would be happy to consider a resubmission, provided the comments of the referees are fully addressed. However please note that this is not a provisional acceptance.

Please find below the comments made by the referees, not including confidential reports to the Editor, which I hope you will find useful.

- 1) A 'response to referees' document including details of how you have responded to the comments, and the adjustments you have made.
- 2) A clean copy of the manuscript and one with 'tracked changes' indicating your 'response to referees' comments document.
- 3) Line numbers in your main document.
- 4) Please read our data sharing policies to ensure that you meet our requirements <https://royalsociety.org/journals/authors/author-guidelines/#data>.

Sincerely,
Dr Sasha Dall
mailto: proceedingsb@royalsociety.org

Reviewer(s)' Comments to Author:

Referee: 2

Comments to the Author(s).

The initial, background section nicely points out why the dynamics of change in learned song are likely to differ from, and relate to, genetic change, under selection. The paper describes important

and original insights, capitalizing on data on territorial/mate-attracting song in a group of birds that can be used to investigate the relationship between learned and evolutionary-genetic changes in song. Looking at these questions with an overarching phylogenetic view, while relating it to the more usual micro-evolutionary perspective, and using a particular example (sunbirds) is mind-expanding for anyone interested in the evolution of social signals, especially those influenced by learning. Given the complexity of the questions, this paper manages to clearly work through the many conceptual complications to an admirable degree.

The one very small clarifying suggestion I have is that it should be pointed out as soon as the word "territorial" is used that the calls that are the focus of this study function as both territorial and mate-attraction signals. Many of the points discussed, such as relevance of song to reproductive isolation and speciation, usually refer to courtship song and female choice, which has different dynamics than do threat and exclusively territorial (e.g., aggressive) signals. [Due to the genetic correlation between signals and choice in courtship signaling the evolution of courtship songs are expected to change more rapidly and attain greater complexity than territorial signals, which are threat signals whose dynamics include imposition of "truth" by ability to back up threats with real strength.] Given the importance of this distinction the place to point this out is on line 106: change that sentence to state that the paper focuses on "learned territorial/mate attracting songs" of sunbirds.

Small suggestions:

Ln 37 - "dynamics" might be better than "trajectories"

40 - longer should be "long"

Ln 116 say in a parenthesis what "evolutionary mode" means, e.g. (punctuation, stasis, gradual change)

122 change "of" to for, or regarding

131 - can you change "multifarious" to "not uniform?" Or just variable or "varies in its nature," which better relates to the next sentence.

Referee: 3

Comments to the Author(s).

The African sunbirds appear to be an excellent system to study song evolution, given the six (or so) species and separated populations within each species, along with the presence of some contact zones. The authors of this manuscript have collected an impressive number of song recordings (419 songs from 142 individuals), setting them up for a reasonably comprehensive survey of song variation within and among the 6 species. This manuscript uses a mapping of song variation onto a molecular phylogeny to conclude that a model with bursts of evolution fits the song variation better than a model of random change at a constant expected rate (Brownian motion). The authors' interpretation of this result is that songs show a "punctuated evolution" pattern of song variation, and that songs have experienced long periods (" $> 10^6$ years") of near stasis.

Unfortunately, this conclusion is not convincing given the analyses and data presented. The main signal for the "burst" of evolution comes from the comparison of the fuelleborni species compared to the others. For instance, Fig. 2 shows that the 5 populations of fuelleborni have similarly long songs compared to most other species in the phylogeny, and so a comparison of the two models indicates support for the burst model over the Brownian motion model. There is however an alternative explanation that has not been sufficiently considered by the manuscript: Movement of individuals between the different fuelleborni populations could have spread a similar song type between those populations. This could happen as a result either of genetic exchange or purely memetic (transmission of traits through learning) exchange, and could be a result of selection (a new song type is advantageous) or mutation and drift alone. It is not clear why the manuscript concludes from the similarity of song in these populations that songs have been in stasis for more than a million years. Much of the reasoning appears to be based on the molecular phylogeny suggesting the populations started diverging that long ago, but the genetic

data were analyzed with a method that can only present the data in a bifurcating phylogeny, rather than methods that have the capability to indicate genetic or memetic exchange between populations.

Another reason that stasis is not very convincing is that there is in fact much within-species variation in song. This is mentioned in the text, and is visible in Figure A1. The bivariate plots in that figure show much variation within species, and much overlap between clouds of points for different species. In particular *feulleborni* shows much variation, and overlap or contact in acoustic space with other species.

The analysis considers two models of song variation, and it is worth considering whether there are others that may be more consistent with the data and the evolutionary process. A model that was not considered is one in which there is Brownian motion but the rate of that Brownian motion can itself change gradually through the phylogeny. This is likely more consistent with what is known about song evolution. If that is truly how songs are evolving, then the constant-rate Brownian motion model would not be well supported, and the alternative in this manuscript of a burst model might then appear to be favored in comparison. In other words, when only comparing two models, one might look better than the other but they might both be far from the truth.

Although the manuscript is about song evolution, the manuscript is actually not very clear about the pattern of variation in song. It is difficult to see details and compare songs in the Fig. 2C, and showing just a single song for each species seems insufficient given the apparent variation in songs as indicated in Fig. A1. It seems that examining and somehow presenting a summary of song variation within individuals, between individuals within a population, and among populations is important for the reader to understand the overall context of song variation, and only then can hypotheses regarding the evolutionary process be effectively addressed.

A few other recommendations:

It would be good if the different figures were more integrated in terms of terminology, use of color and symbols, etc. For example, it is presently unclear whether the *mediocris* N vs. S in Fig. 1 corresponds to *MtKenya* vs. *Mbulu* in Fig. 2. The colors used for taxa in Fig. A1 are different than those used in Fig. 1.

There are a variety of phylogenies used in the main paper and the appendix and supplement, and the captions often don't give many details about the source of the data and the method that built the phylogeny. This makes it difficult for the reader to understand the different appearance of the different phylogenies (e.g. compare Fig. A2, where variation among most pops of *feulleborni* seems very low; vs. Fig. 2, where it looks much deeper).

The paper mostly presents univariate analyses of the different song variables—it says that a multivariate clustering approach generated distinct clusters, but there doesn't seem to be a figure showing those clusters in a way that the reader can evaluate the support for them (e.g., how distinct were they, how much overlap). It would be good if some sort of figure such as a PCA or similar were shown.

Overall, the authors appear to have a valuable dataset on song variation in this interesting group of birds. A paper that presents the patterns of variation more clearly to the reader and that considers other models than just the two here would be of much value. In particular, it would be good if the authors considered (1) the possibility of the present song variants having spread among populations within a species rather recently, rather than evolving strictly through a bifurcating phylogeny, and (2) the possibility that neither the constant-rate Brownian motion nor the burst model being the best models of real song evolution.

Referee: 4

Comments to the Author(s).

General comments:

This study introduces a novel technique for measuring pulsed phenotypic evolution, and applies it to a fascinating empirical system: the evolution of culturally evolving song in an 'land island' species complex. The paper is therefore ambitious, and of potentially broad impact on two fronts.

(1) As someone with only shallow understanding of comparative phylogenetic methods, I found it difficult to assess the new technique for measuring pulsed evolution. It seems to me that the new method could easily have been presented as a standalone paper, and relegating it to a supplemental materials has perhaps meant that less effort has gone into making it easy to follow for interested readers with a limited expertise, like me. I would have appreciated, for example, an explanation of how this approach compares to related (I think) methods examining different evolutionary rates within a phylogenetic tree by O'Meara et al, Revell etc. I also think that more could have been done to explain the figures, especially S1, S2, S6 and S7 - even as far as what high values of var_d and var_p imply. Finally, since this is the introduction of a brand new method in a field where previous studies have been controversial, I think that the paper would have benefitted from efforts to examine its performance with simulated data. I realise that you have simulated data to correct for AICc, but would it make sense to go one step further and demonstrate that this correction is itself reliable, by applying the whole analysis pipeline to simulated data? If given phenotypic data simulated under Brownian evolution, what is the probability that the method (in toto) would erroneously detect pulsed evolution?

(2) The paper examines phenotypic evolution of song over a molecular phylogenetic tree of a species complex. I did not find any discussion of potential gene flow between populations - even though this is a closely related group of populations. I think this is potentially problematic for the statistical inference of pulsed phenotypic evolution. Gene flow would, of course, tend to homogenise populations in contact, and give an appearance of comparatively ancient divergence between them and populations not in contact. In other words: gene flow could generate an illusion where a recent selective sweep across populations would appear as rapid evolution before the MRCA. In fact, one example of such gene flow is explicitly mentioned in the Discussion (Lines 258-260)

Tests for gene flow should be easy to carry out, and my concern easily dealt with. In this particular case, the situation is potentially more tricky because song might spread between population boundaries even when genes don't - through cultural evolution.

An alternative explanation for the pattern of phenotypic divergence that is described is that song is regularly being shared across population boundaries, and that geographic distance is therefore a better predictor of song similarity than phylogenetic distance. It would be great if the authors could test and refute that alternative. I realise that there is some possible evidence that songs may not pass across one population boundary, but can that be applied to the critical boundaries *within* the fuelleborni clade, where most of the pulsed evolution seems to occur?

(3) At some points, I felt that the manuscript conflated the acoustic measurements that were made with the cultural units that were transmitted from one individual to another. Individual song-types or syllables are learned by birds; in songs like these, they are very high-dimensional trajectories of how frequency changes over time. Measurements such as "mean peak frequency" obviously only capture a small proportion of that variation: I'm sure there are many different song-types in these populations that share the same mean peak frequency. Many of the 14 acoustic features that were measured are inter-correlated, and together, I would bet that they only capture a small proportion of the variation in song structure.

This impinges on the interpretations in the manuscript in several ways. First of all, broad-scale acoustic measurements, like those used here, are likely to reflect general constraints acting on

learning (caused by vocal tract morphology, environmental selection, or perceptual predispositions). I would argue that it's the norm in other species that such measurements are stable over large periods of time. For example, studies of several species have explicitly demonstrated that general song features are constant over continent-wide scale populations of millions of individuals (e.g. black-capped chickadees, chaffinches, chiffchaffs, swamp sparrows, just to name a few). This geographic consistency implies considerable stasis over time too.

So in the Introduction and the Discussion (e.g. L246-249), it feels to me that the manuscript constructs straw men. There should have been clear a priori expectations that song features such as duration and peak frequency show stasis over long periods of time *because* these likely reflect stable constraints on learning, rather than cultural evolution itself, and *because* evidence from other species shows considerable stability over large distances.

(4) Learned bird songs are typically thought to develop through a combination of genetically transmitted constraints (in terms of what is perceived as conspecific, and what can be produced with the vocal organs) and cultural transmission. It is not immediately clear what these results suggest in this regard. It could be the case that the pulses of change represent evolution in underlying constraints. The manuscript is somewhat unclear about this - it acknowledges that predispositions and constraints exist, but focuses on the learned aspect of song, but then brings up predispositions again in the conclusions. I would personally prefer it if the manuscript was a little more explicit.

(5) I think the manuscript neglects literature on learned bird song and speciation. This applies both to modelling studies (by Servedio and colleagues), but also to other studies of island diversification and bird song.

Specific Comments

Line 72 - nb - I don't think that the reference cited here reports original empirical findings. I am not aware of any strong evidence that novel songs are at a selective advantage (although this has frequently been proposed)

Line 82 - quite a lot of empirical data suggests an alternative pattern, however: that the "acoustic space" permitted by genetic predispositions becomes saturated within even moderately sized populations, such that little differentiation is observed between isolated populations. It feels somewhat as if you are setting up a straw man here.

Lines 84 - but explicit models of the role of song learning in speciation are not cited. This includes Lachlan & Servedio (which is cited only in the Conclusions), but also other modelling studies by Servedio.

Lines 92-98 - It seems to me that - as fantastic a study as it is - there is rather too much space taken with discussing the white-throated sparrow study. There are plenty of other studies showing patterns relevant to this one, especially long-term stasis (e.g. Pipek et al in yellowhammers, Lachlan et al. in swamp sparrows).

Lines 87-95 - These two examples seem to be confusing two different levels of variation. In greenish warblers, broad scale acoustic features were measured - such as minimum frequency. These measures might be expected to reflect broad constraints on song variation, but not to identify individual song-types.

In contrast, the white-throated sparrow study looks at a particular song-type - albeit with very constrained species levels of variation.

Another key example of pulsed variation is humpback whale song (e.g. Garland et al 2011, Curr. Biol.)

Lines 171-179. Your statistical analysis did not seem to take into account any correction for multiple comparisons. However, you are comparing 14 song traits (some which are likely to be inter-correlated), without any a priori hypotheses for which should show pulsed evolution. Therefore, it would seem appropriate to me that your statistical discovery of pulses should take into account the risk of an inflated type II error. If you were to simulate 14 phenotypic traits under Brownian phenotypic evolution, what would be the probability of finding 4 show clear signs of pulsed evolution?

Line 185 - It would be useful to provide a composite tree in the main paper showing the placement of pulses across different traits.

Line 205. Studies of stasis in learned bird song have been carried out: Pipek et al on yellowhammers; Lachlan et al. 2018 on swamp sparrows, among them.

Line 312. I think this is clumsily phrased: there is no empirical whatsoever for genetic assimilation of song traits. The sources mentioned write about genetic assimilation much more generally.

Supplemental Information

At the foot of P6 there must be a typo /missing words in the sentence beginning "Informally..." Please check throughout for typos - in some places I found it hard to follow I think because of similar typos.

Table S2 - I found this confusing. Delta A is used both to refer to the four CI columns, but then there is a separate Delta A column, which is not explained.

Fig. S5 - the axis of these figures is not described, nor the units.

Fig. S8 - the scale bar is not explained. The figure legend should carefully explain what the thick bar means, and what it was based on. After reading the supplemental materials carefully, I can work most of it out, but the figure legend should make it more explicit.

Author's Response to Decision Letter for (RSPB-2021-0498.R0)

See Appendix B.

RSPB-2021-2062.R0

Review form: Reviewer 2

Recommendation

Accept as is

Scientific importance: Is the manuscript an original and important contribution to its field?

Excellent

General interest: Is the paper of sufficient general interest?

Excellent

Quality of the paper: Is the overall quality of the paper suitable?

Excellent

Is the length of the paper justified?

Yes

Should the paper be seen by a specialist statistical reviewer?

No

Do you have any concerns about statistical analyses in this paper? If so, please specify them explicitly in your report.

No

It is a condition of publication that authors make their supporting data, code and materials available - either as supplementary material or hosted in an external repository. Please rate, if applicable, the supporting data on the following criteria.

Is it accessible?

Yes

Is it clear?

Yes

Is it adequate?

N/A

Do you have any ethical concerns with this paper?

No

Comments to the Author

All criticisms on the former submission have been satisfactorily met.

Review form: Reviewer 4

Recommendation

Accept with minor revision (please list in comments)

Scientific importance: Is the manuscript an original and important contribution to its field?

Good

General interest: Is the paper of sufficient general interest?

Excellent

Quality of the paper: Is the overall quality of the paper suitable?

Acceptable

Is the length of the paper justified?

Yes

Should the paper be seen by a specialist statistical reviewer?

Yes

Do you have any concerns about statistical analyses in this paper? If so, please specify them explicitly in your report.

Yes

It is a condition of publication that authors make their supporting data, code and materials available - either as supplementary material or hosted in an external repository. Please rate, if applicable, the supporting data on the following criteria.

Is it accessible?

Yes

Is it clear?

Yes

Is it adequate?

Yes

Do you have any ethical concerns with this paper?

No

Comments to the Author

I found the manuscript considerably clearer and more consistent than the previous version - and still of considerable interest. I appreciate the work that has gone into clarifying the methods and interpretation, which has allowed me to understand the paper more clearly than in my initial review.

I have one significant concern remaining. The emphasis of the paper seems very clear now: that although previous work may have argued that bird song evolved in a gradual way, that is not the case here. Instead, songs remain in stasis for long periods of time, and then undergo change in bursts of punctuated evolution.

The evidence for this comes from the contrast of two “idealised” models of evolution: Brownian motion versus punctuated evolution. In the punctuated evolution model, evolution, the variable X_k is set to 0 on all but up to 6 nodes. In the SI, this simplified approach is justified as: “because we are interested in detecting rather than precisely quantifying, punctuated evolution...”.

However, there is a strong emphasis in the paper on the stasis between the punctuated episodes too. I have to admit that I do not really see how evidence *for* punctuated equilibrium precludes gradual evolution between bursts. Because you don’t include a model with both Brownian and punctuated evolution possible (as in Landis and Schraiber), I don’t see how you can really test the idea of stasis. Surely all you can say is that, for those four traits, your data fits more closely to the punctuated evolution model (with complete stasis in between bursts) than to one with one fixed gradual rate.

I realise that this is closely related to a comment one of the other reviewers made previously, and that you introduced the LMM, partly in response. But the LMM suggests that some of the variation in traits is accounted for between-individual, within-population variance. It might be a relatively low proportion of the evolutionary change occurring in this clade, but (perhaps except for duration) it seems to be clearly >0 . Doesn’t that suggest a combination of slow gradual evolution interspersed with rapid punctuated bursts? If so, then the question becomes: is the slow gradual evolution slow enough to justify the conclusion that song traits are in “stasis” for hundreds of thousands of years?

Specifically, I don’t see how several statements in the paper can be supported with the methods employed:

LL331-334: what is the evidence that evolution between pulses must be “minimal”?

L344: you say that between jumps, evolution must either be in stasis or “highly bounded”. I’m not completely sure what you mean by highly bounded here. Does it allow for gradual evolution?

L406: here again, evolution must be “highly bounded, approaching stasis” suggests that your method allows you to establish an upper limit of the rate of gradual evolution between bursts. If it does, I don’t think this has been explained fully.

L664: “Strongly bounded evolution (stasis or near-stasis)” - same point again. I’m not sure what “near-stasis” really means - in particular in relation to the biological context of the paper. Does near-stasis entail “similar or slower rate of evolution to that found for non-learned traits”? If not, then should your conclusions re song learning be modified somewhat?

In a nutshell: can you rule out the possibility that, compared to a non-learned trait, gradual evolution between bursts is actually rapid, but that it’s just much more rapid still during the bursts themselves? If so, that needs to be presented much more clearly to the reader. If not, then I think you need to rethink your interpretation of the results.

Minor points:

L103: I think this is the first use of the term “highly bounded”. It would be good to explain exactly what you mean by this. I think you just mean “low rates of evolution”. It could simply mean: “with very clearly delimited upper and lower limits of evolutionary rate”. But I think that most papers have used bounded evolution to refer to the restriction of the evolution of continuous characters to phenotypic limits. I’m not sure how this applies to your analysis or data, however.

L224: here or in the Table legend, it might be worth explaining that the residual variation refers to between-song-type but within-individual variation (if I’ve understood correctly).

L227-228: in every species of songbird I can think of, individual-level variation would be much higher than population-level variation. The reason why people have argued that learned bird song might evolve rapidly is because the exceptionally high within-population variation can drive rapid divergence between populations. But that implies a very high level of variation between learned song types within a population that is still often much larger than between-population variation. I don’t quite follow the logic for why your results mean that geographic variation between populations must be “minimal”.

In addition - I don’t think there is any description of how individuals were sampled within populations. Most species of birds show considerable microgeographic variation over a range of a few km. Were individuals sampled across the population range, or mostly from one location?

L352 - typo/missing word here.

Methods / SI - I do think you need to explain why you didn’t use an established method like Landis & Schraiber to test for pulsed evolution. The explanation on line SI79 was not very clear to me in this respect.

An SI table would be very useful to provide fuller details on sample sizes from different populations, recording locations, sources of recordings, songs analysed per population etc.

Decision letter (RSPB-2021-2062.R0)

18-Oct-2021

Dear Dr McEntee

I am pleased to inform you that your manuscript RSPB-2021-2062 entitled "Punctuated evolution in the learned songs of African sunbirds" has been accepted for publication in Proceedings B.

The referee(s) have recommended publication, but also suggest some minor revisions to your manuscript. Therefore, I invite you to respond to the referee(s)' comments and revise your manuscript. Because the schedule for publication is very tight, it is a condition of publication that you submit the revised version of your manuscript within 7 days. If you do not think you will be able to meet this date please let us know.

[http://datadryad.org/submit?journalID=RSPB&manu=\(Document not available\)](http://datadryad.org/submit?journalID=RSPB&manu=(Document%20not%20available)) which will take you to your unique entry in the Dryad repository. If you have already submitted your data to dryad you can make any necessary revisions to your dataset by following the above link. Please see <https://royalsociety.org/journals/ethics-policies/data-sharing-mining/> for more details.

Sincerely,
Dr Sasha Dall
mailto:proceedingsb@royalsociety.org

Reviewer(s)' Comments to Author:

Referee: 4

Comments to the Author(s).

I found the manuscript considerably clearer and more consistent than the previous version - and still of considerable interest. I appreciate the work that has gone into clarifying the methods and interpretation, which has allowed me to understand the paper more clearly than in my initial review.

I have one significant concern remaining. The emphasis of the paper seems very clear now: that although previous work may have argued that bird song evolved in a gradual way, that is not the case here. Instead, songs remain in stasis for long periods of time, and then undergo change in bursts of punctuated evolution.

The evidence for this comes from the contrast of two "idealised" models of evolution: Brownian motion versus punctuated evolution. In the punctuated evolution model, evolution, the variable X_k is set to 0 on all but up to 6 nodes. In the SI, this simplified approach is justified as: "because we are interested in detecting rather than precisely quantifying, punctuated evolution...".

However, there is a strong emphasis in the paper on the stasis between the punctuated episodes too. I have to admit that I do not really see how evidence *for* punctuated equilibrium precludes gradual evolution between bursts. Because you don't include a model with both Brownian and punctuated evolution possible (as in Landis and Schraiber), I don't see how you can really test the idea of stasis. Surely all you can say is that, for those four traits, your data fits more closely to the

punctuated evolution model (with complete stasis in between bursts) than to one with one fixed gradual rate.

I realise that this is closely related to a comment one of the other reviewers made previously, and that you introduced the LMM, partly in response. But the LMM suggests that some of the variation in traits is accounted for between-individual, within-population variance. It might be a relatively low proportion of the evolutionary change occurring in this clade, but (perhaps except for duration) it seems to be clearly >0 . Doesn't that suggest a combination of slow gradual evolution interspersed with rapid punctuated bursts? If so, then the question becomes: is the slow gradual evolution slow enough to justify the conclusion that song traits are in "stasis" for hundreds of thousands of years?

Specifically, I don't see how several statements in the paper can be supported with the methods employed:

LL331-334: what is the evidence that evolution between pulses must be "minimal"?

L344: you say that between jumps, evolution must either be in stasis or "highly bounded". I'm not completely sure what you mean by highly bounded here. Does it allow for gradual evolution?

L406: here again, evolution must be "highly bounded, approaching stasis" suggests that your method allows you to establish an upper limit of the rate of gradual evolution between bursts. If it does, I don't think this has been explained fully.

L664: "Strongly bounded evolution (stasis or near-stasis)" - same point again. I'm not sure what "near-stasis" really means - in particular in relation to the biological context of the paper. Does near-stasis entail "similar or slower rate of evolution to that found for non-learned traits"? If not, then should your conclusions re song learning be modified somewhat?

In a nutshell: can you rule out the possibility that, compared to a non-learned trait, gradual evolution between bursts is actually rapid, but that it's just much more rapid still during the bursts themselves? If so, that needs to be presented much more clearly to the reader. If not, then I think you need to rethink your interpretation of the results.

Minor points:

L103: I think this is the first use of the term "highly bounded". It would be good to explain exactly what you mean by this. I think you just mean "low rates of evolution". It could simply mean: "with very clearly delimited upper and lower limits of evolutionary rate". But I think that most papers have used bounded evolution to refer to the restriction of the evolution of continuous characters to phenotypic limits. I'm not sure how this applies to your analysis or data, however.

L224: here or in the Table legend, it might be worth explaining that the residual variation refers to between-song-type but within-individual variation (if I've understood correctly).

L227-228: in every species of songbird I can think of, individual-level variation would be much higher than population-level variation. The reason why people have argued that learned bird song might evolve rapidly is because the exceptionally high within-population variation can drive rapid divergence between populations. But that implies a very high level of variation between learned song types within a population that is still often much larger than between-population variation. I don't quite follow the logic for why your results mean that geographic variation between populations must be "minimal".

In addition - I don't think there is any description of how individuals were sampled within populations. Most species of birds show considerable microgeographic variation over a range of a few km. Were individuals sampled across the population range, or mostly from one location?

L352 - typo/missing word here.

Methods / SI - I do think you need to explain why you didn't use an established method like Landis & Schraiber to test for pulsed evolution. The explanation on line SI79 was not very clear to me in this respect.

An SI table would be very useful to provide fuller details on sample sizes from different populations, recording locations, sources of recordings, songs analysed per population etc.

Referee: 2

Comments to the Author(s).

All criticisms on the former submission have been satisfactorily met.

Author's Response to Decision Letter for (RSPB-2021-2062.R0)

See Appendix C.

Decision letter (RSPB-2021-2062.R1)

26-Oct-2021

Dear Dr McEntee

I am pleased to inform you that your manuscript entitled "Punctuated evolution in the learned songs of African sunbirds" has been accepted for publication in Proceedings B.

Data Accessibility section

Open Access

Paper charges

Sincerely,
Editor, Proceedings B
mailto: proceedingsb@royalsociety.org

Appendix A

Missouri State
U N I V E R S I T Y

February 21, 2021

Dear Dr. Dall and Referees,

Thank you very much for your feedback on the manuscript. I am resubmitting this manuscript, RSPB-2020-1559, entitled "Punctuated evolution in the learned songs of African sunbirds", for publication in Proceedings of the Royal Society B.

The focus of the suggested revisions were on the writing of the introduction and discussion, especially streamlining the introduction, and balancing the discussion of evidence for gradual and punctuated evolution patterns in both the introduction and discussion. In general, we have approached revisions for the introduction by removing parts that raised questions that we did not address in our analyses, and then substantially editing the introduction to explain some evidence for gradual and punctuated evolution in songs of song-learning birds.

Then, in the discussion, we have added a paragraph devoted to discussing the variation among traits in support for gradual and punctuated evolution. Further, we have attempted to clarify the parts of the discussion regarding the potential role of geographic range changes in pulses of trait evolution, per suggestions from the Referees. And finally, we have modified language in the Discussion where Referees found statements to be misleading, confusing, or to go too far.

Regarding the potential for conformity bias in learning to stabilize song traits, we argue that this mechanism is less important for the spectrotemporal traits we've measured, and to our knowledge has not been shown to plausibly operate to stabilize traits on the long timescale we're faced with in our data. We provide more detail on this point in our response to Referee 1.

Please see responses to the Editor's and Referees' comments below. Line numbers refer to lines in the "tracked changes" document. My responses below are in colored text so they are more easily distinguished.

Sincerely,

Jay McEntee
Missouri State University

DEPARTMENT OF BIOLOGY

901 South National Avenue, Springfield, MO 65897

jaymcentee@missouristate.edu

An Equal Opportunity/Affirmative Action/Minority/Female/Veterans/Disability/Sexual Orientation/Gender Identity Employer and Institution

21-Aug-2020

Dear Dr McEntee:

I am writing to inform you that your manuscript RSPB-2020-1559 entitled "Punctuated evolution in the learned songs of African sunbirds" has, in its current form, been rejected for publication in Proceedings B.

This action has been taken on the advice of referees, who have recommended that substantial revisions are necessary. With this in mind we would be happy to consider a resubmission, provided the comments of the referees are fully addressed. However please note that this is not a provisional acceptance.

Sincerely,

Dr Sasha Dall
mailto: proceedingsb@royalsociety.org

Associate Editor
Board Member: 1
Comments to Author:

I agree with the reviewers that the results of this study are intriguing and the model system is great for addressing evolutionary patterns of learned traits. Both reviewers have excellent insights, all of which should be addressed. In particular, both are looking for more in-depth discussions of the role of geographic range changes and the role of learning in this process. Authors also should modify the introduction and discussion as

suggested, including opportunities for clarification and a more balanced discussion of evidence for both gradual and punctuated evolutionary patterns.

Reviewer(s)' Comments to Author:

Referee: 1

Comments to the Author(s)

Highlights

Past work has focused on the role of learned song as a pre-mating isolating mechanism and fewer studies have asked about the tempo and mode of song evolution. Those that have focused on gradual evolution. The authors use a fascinating songbird system to ask whether gradual or punctuated evolution best describes the evolution of song traits. Overall, this is an interesting idea and the methods appear sound to me (a few comments below). However, I think that the authors need to scale back the introduction and the discussion to align with the scope of the methods and results. I also think there is a strong emphasis on discussing those song traits which showed punctuated evolution, when these were actually in the minority. This is not to say that these traits aren't interesting – evidence of punctuated evolution is quite interesting – but I would make sure to discuss the evidence for gradual evolution as well.

Major comments

Introduction is very thorough, which is great, but reads more like a review (which I think is actually a much needed review, and one the authors should consider separately perhaps?). I suggest paring the introduction back to focus on setting up the specific points that are tested in this manuscript.

JPM: We have removed two paragraphs that introduced additional points that are not tested in the manuscript (lines 78 and 86 in the .docx with tracked changes).

I think the final paragraph could be better focused with a clear hypo-deductive framework. There are statements about methods (multivariate and univariate approaches) but it isn't clear how these approaches link to a question set up earlier in the intro. Then the final paragraph turns to a point about gradual versus punctuated evolution. This makes it sound like this is the main and only question tested in this study. Although there is a nice set up for why one might expect gradual evolution, the set up for punctuated evolution is not as clear. This is particularly surprising given the general depth of the introduction overall. When I read the final paragraph, I was left wondering whether the study was going to ask any of the other questions brought up earlier in the introduction. I was also left wondering how the referenced methods (multivariate and univariate) specifically mapped onto testing gradual versus punctuated evolution.

JPM: We have expanded the parts of the introduction discussing punctuated evolution in learned song, such that the reasoning for comparing between gradual and punctuated evolution should be clearer. We make statements on our approach to examine evidence for distinct phenotypic clusters in multivariate space here to make clear that the end result of the evolutionary process is a set of distinct songs that map well onto monophyletic clades in our evolutionary trees (lines 198-203 in tracked changes .docx). Our evolutionary modeling approach is univariate, thus we made the distinction in this paragraph initially because we were performing analyses on each of the traits we measured.

In your discussion of the sources of stabilizing selection, you mention male and female receivers. The way in which this is discussed makes it sound like selection is occurring in the contexts of mate choice and male-male competition. Another source of stabilizing selection on these cultural traits is the learning process itself. The mode of song learning could be strongly stabilizing if there is conformity bias in the learning process (see papers by Rob Lachlan on this point).

JPM: We agree conformity bias could stabilize aspects of phenotypes over periods of time that are interesting for cultural traits like syllables (close to our “elements”). However, because we have measured spectrotemporal traits of songs, and not the syllables that are culturally transmitted more as memes, we do not think conformity bias is as relevant as an explanation. Moreover, we estimate that periods of stasis in our measured spectrotemporal traits are on the order of hundreds of thousands of years or more. Lachlan et al. 2018 describe conformity bias as allowing for traditions to persist over 500 years. Such conformity bias could play a role, but we argue that it would not be enough over the timeframe we are dealing with.

There is an interesting suggestion that range shifts could drive punctuated evolution, but, unless I’m mistaken, there is on average one shift in song traits whereas I would imagine there would be more range shifts over time. How might the placement of shifts in song evolution tell us what might underly these shifts?

JPM: We have added two sentences in the discussion to clarify (lines 438-441 in tracked changes .docx). Range expansions are indeed likely to be much more frequent than evolutionary pulses, as they may make the conditions for evolutionary change more likely, but do not guarantee it will happen (e.g. Slatkin 1996 on founder-flush models, now cited in the text).

One component missing is a discussion of those song traits that exhibit gradual evolution. What of those? Why might some song traits show punctuated but most show gradual evolution? How might that comparison give insight into what might be driving punctuated evolution?

JPM: We’ve added in a paragraph to discuss this point, which focuses on a possible association between punctuated evolution and those traits that differ most across the multivariate phenotypic clusters (i.e. ‘species’, but where the southern populations of mediocris are treated separately from the northern populations). The traits with more relative support for gradual evolution do not differ by as much among species, on average (lines 322-331).

Minor comments

Line 194 – the text here suggests that Fig 2 will illustrate the degree to which pulses are co-localized, but it appears that only song duration is plotted in figure 2. What is missing?

JPM: We understand the confusion here. We were attempting to direct readers to the Appendix and main text both to compare the figures, as we anticipate that space will prevent us from including more than one example in the main text. Instead of including more of these figures in the main text, we have included a verbal description here of the extent of co-localization of pulses (lines 292-293).

Line 217-291 This sentence is awkwardly phrased and a bit hard to follow as such.

JPM: Rephrased for simplicity (lines 334-335).

Line 253 – 256 This is another sentence that is quite densely worded and thus hard to follow. Perhaps unpack into multiple, simpler sentences?

JPM: Indeed! Separated into two sentences (lines 384-385).

Line 266 – At this point, I realized that you hadn't actually mentioned the song traits exhibiting punctuated evolution – are these temporal or spectral traits or both? Would help to know as you start to discuss potential mechanisms.

JPM: Sentence inserted in the Discussion's first paragraph to clarify that these are both temporal and spectral traits (line 312-313). Additionally, we have included a verbal explanation of the traits corresponding to the differing levels of support for punctuated versus gradual evolution.

Line 283 – 286 – I see your reasoning in here but all of this is conjecture as you don't include any analyses that actually test how morphological traits are evolving or for correlations among morphological and song traits. I think your hypothesis is highly plausible, particularly as morphological traits like body size are much less of a constraint on frequency characters in songbirds (which control frequency using syringeal muscles) than in suboscines, for example, which are more likely to show correlated evolution of bod size and peak frequency, but I think you have to be careful not to conjecture too far, particularly without analyses to support these points.

JPM: Point taken here. Though we feel that the original statements are justified, we have not done a formal analysis within the manuscript, as the reviewer notes. We have revised the end of this paragraph (lines 418-420).

Line 302 – The traits showing pulsed evolution are mentioned here – I would make sure to state these clearly in the results section.

JPM: These are now stated in the results section (lines 278-283).

Line 303 – I agree that studies have found evidence of underlying genetic variation for fine-scale temporal and duration characters. I am not convinced by frequency, and which frequency trait do you mean here? All frequency traits?

JPM: For the evidence for genetic variation underlying frequency, we have cited Munding and Lahti (2014), who used a crossing experiment with different genetic lines of canaries to show that genetic identity predicted proportions of high-pitched versus low-pitched syllables. We have edited this sentence so that we are not referring to specific traits, as a full discussion of the genetic evidence would take up too much space (lines 449-451).

Line 325 – Can you be more specific by what you mean here regarding 'discontinuities in evolutionary processes'?

JPM: We have removed this phrase, and attempted to clarify meaning here (lines 476-477).

Line 331 – I would be careful with this first sentence as there is substantial work on this idea, beginning with the work of Cavalli-Sforza and Feldman in humans, that provides quite a bit of insight into how learning shapes evolutionary processes. I agree that less is known in birds.

JPM: We have modified this statement to read: "The effects of learning on evolutionary diversification processes are poorly explored for many organisms." (lines 481-482).

Methods

Overall, the methods look spot on to me. Luscinia is a great program to use, especially for field recordings that have low signal to noise ratios. I would make sure to make a statement about how you dealt with common issues of taking frequency measurements (e.g., signal amplitude and signal distortion effects on frequency measures – see (1))

References

1. S. A. Zollinger, J. Podos, E. Nemeth, F. Goller, H. Brumm, On the relationship between, and measurement of, amplitude and frequency in birdsong. *Animal Behaviour* 84, e1-e9 (2012).

JPM: Our approach to frequency measurements uses the extraction of peak frequencies from amplitude spectra, and is therefore not subject to the issues raised by Zollinger et al. 2012, which arise from manually selecting frequency windows and treating these manual selection as minimum and maximum frequencies of the signal. We have added in a sentence to clarify that our approach should not have these issues (lines 548-550).

Referee: 2

Comments to the Author(s)

These comments are in order of line number, not importance. Some important conceptual comments are at the end. I have no comments on the phylogenetic methods which are outside of my expertise.

55- “long been thought” might cite Mayr 1963 book on *Animal Species and Evolution*
JPM: Citation inserted.

58 – Mayr there too, on reproductive isolation.
JPM: Citation inserted.

68 “learned signals are potentially subject to different evolutionary pressures” - Need to make more precise statements than this since “evolutionary pressures” is meaningless by itself. What you could mean and should say is that learned signals (learned phenotypes) are subject to non-genetic variation. As written a reader might think that selection operates differently on them (people like to refer, vaguely, to “selection pressure”), which would not be true. Selection sees phenotypes, including learned ones, without any regard to underlying mechanism of variation, whether learning or genetic.

JPM: We have edited this line to be more specific about which evolutionary pressures and trajectories we have in mind (lines 66-74). “Thus, learned signals are subject to different evolutionary pressures, including the introduction of culturally transmitted novelties and cultural drift (1, 6), and may exhibit different evolutionary rates or trajectories (e.g. extents of gradualism versus punctuated evolution) over time (11), as compared to signals that are not learned.”

68 “...may exhibit different evolutionary trajectories” - similarly vague. Say how you think they might be distinctive! If you don't have a specific meaning in mind then leave this out – giving a reference does not clarify the sentence (the reader should not have to read another paper to find out).

JPM: See response to previous comment.

70-71 "...cultural novelties appear frequently," This would make sense if it said "cultural (learned) novelties may be repeated frequently and therefore have their potentially complex properties be subject to selection frequently compared to those, eg. mutationally-mediated traits, that require positive selection to spread and have their complex features subject to selection." If you mean something else please state it precisely.

JPM: Here we meant to say first that the rate of introduction of novel phenotypes (in processes analogous to mutation) can be high in learned traits, and we have altered this point in the text, citing Lynch (1996) (lines 75-77). For the second point, about novelty potentially being advantageous, we cite West-Eberhard (1983).

79 – "specific conditions" such as what? Again, the thought starts off well but becomes vague in the end. The reader has to keep guessing at what your point is supposed to be.

JPM: We have removed this paragraph to make room in the introduction for a discussion of evidence for gradual versus punctuated evolution in bird song specifically, as the paragraph that contained this point brought up a number of questions that we cannot address with our data.

87 – species don't learn. Do you mean that individuals of a particular species have species-specific limits on what they learn?

JPM: Corrected (lines 81-83).

132 ff – can you say "pattern" or even "phenotypic pattern" rather than "mode?" I realize that tempo and mode were successfully used by G. G. Simpson, e.g. in a book title, but these days "mode" could mean drift vs selection. Aren't you going to be looking for patterns and even recurrent patterns of phenotypic change? Again – please give the reader a concrete image of what you are trying to say.

Response: We believe this phrasing ("tempo and mode") has become so standard in phylogenetic approaches examining similar questions to ours (regarding the temporal pattern of evolution) that mode carries more meaning for most readers than pattern in this context. We are specifically interested in investigating the time course of evolutionary divergence, and we have tried to make this evident throughout.

136 – maybe this question has not been put into the dichotomy you pose here, but bird song and plumage diversification can be seen in terms of continuous change under sexual selection only leading to diversification when speciation (population isolation, as on an island or a mountain top) is possible, and the dilution of local changes by mobility among sites is curtailed. That is, gradualism can lead to "punctuation" depending on geographic conditions. The "mode" of evolution has not changed, only the setting – you cannot have diversification without isolation of "gene pools" (breeding populations).

JPM: We agree that it is theoretically possible that a signature of punctuated evolution could emerge from processes that would be viewed as gradual over shorter periods of time. Specifically, a geographic cline in a trait can form over a broad contiguous distribution (appearing gradual in space), with geographically and phenotypically intermediate populations subsequently going extinct, and populations at extreme ends

of the cline surviving. Failure to have sampled the intermediate populations when they existed might then lead us to fail to account for this process. However, we argue that if gradual evolution generally predominates over longer periods of time, we should not see stasis in traits among populations and species that are geographically isolated for long periods of time. This stasis pattern from our data is critical to our inference about the relative importance of gradual and punctuated evolution in our measured traits.

Discussion

241 “here was little reason to expect learned songs to be restricted to peaks, because of their high variability” I agree, but for a different reason: “adaptive peaks” imply a zone of the environment where a particular phenotype is adaptive under natural selection. Selection on signals – social or sexual selection – does not follow that model because it concerns success in social competition, not with respect to some aspect of the environment other than the social environment (composed of conspecifics with which individuals interact locally).

JPM: In the initial submission, we used the adaptive landscape without specifying axes, and this may have led to a lack of clarity here on what we’re positing. Here we use the term in the same way it is used by Mendelson et al. 2014, where (in a 2-dimensional version) the x-axis is trait value, but the y-axis represents the response of receivers to signals (instead of population-level fitness as we understand Simpson’s adaptive landscapes to represent). The metaphor can be expanded to additional dimensions. (In Mendelson et al. 2014 Figure 1, the first two axes are trait values, and the third axis is female response in particular). This version of the adaptive landscape allows us to accommodate signaling traits. We have added a sentence to clarify what our visualized axes are (lines 337-339).

An hypothesis to explain a punctuated pattern of signal evolution that is not discussed here (it is not covered by range expansion and founder effect although they might play a role) is that stasis persists in panmictic populations and a burst of evolutionary change occurs when a population becomes isolated from others, e.g. colonizes an island or a mountain top and becomes isolated there. I could not tell from this analysis what the geographic histories of the various lineages was: were the closely related sub-populations (newly formed species) within clades isolates, and the species showing stasis large or old populations without barriers to inbreeding within them for long periods of time? And were the divergent populations geographic isolates, perhaps newer than the populations showing stasis? That would support the general picture of how diversification and speciation looks under sexual or social selection on traits like territorial signals. The adaptive peak analogy does not seem appropriate to me for reasons just explained.

BUT there is an opportunity to use these data to show how a punctuated pattern might occur under different conditions – panmixis (stasis) vs isolation (rapid divergence) – and much literature on island birds etc with a similar pattern might be cited to support it.

JPM: Before beginning this work, we suspected a pattern as described here might emerge, but what we found instead is that several populations that have been isolated for long periods of time (based on genetic data) have extremely similar songs to closely related populations. Our evidence for stasis thus comes from the surprising similarity of populations that have accrued a lot of genetic dissimilarity. In other words, what is keeping traits similar among populations is not gene flow, but something else (we suggest this must include stabilizing selection given all the opportunity for mutations in genetic regions associated with the traits to accrue). The genetic divergence among populations, which we interpret to indicate isolation, is best seen in the manuscript in the deep genetic divergence among populations within *C. fuelleborni* and *C.*

moreaui (shown in the phylogenetic tree in Figure 2). We hope Figure 2 makes evident that populations within *C. fuelleborni*, for example, have had a substantial period of time to evolve differences in mean song duration, but they have not evolved substantial differences despite the opportunity.

Appendix B

September 16, 2021

Dear Sasha Dall,

Please find our responses to reviews interspersed between comments from referees. I have set our responses off by using a different text color (green).

Sincerely,
Jay McEntee

16-Apr-2021

I am writing to inform you that this version of your manuscript RSPB-2021-0498 entitled "Punctuated evolution in the learned songs of African sunbirds" has, in its current form, been rejected for publication in Proceedings B.

This action has been taken on the advice of referees, who have recommended that substantial revisions are necessary. With this in mind we would be happy to consider a resubmission, provided the comments of the referees are fully addressed. However please note that this is not a provisional acceptance.

Please find below the comments made by the referees, not including confidential reports to the Editor, which I hope you will find useful.

- 1) A 'response to referees' document including details of how you have responded to the comments, and the adjustments you have made.
- 2) A clean copy of the manuscript and one with 'tracked changes' indicating your 'response to referees' comments document.
- 3) Line numbers in your main document.
- 4) Please read our data sharing policies to ensure that you meet our requirements <https://royalsociety.org/journals/authors/author-guidelines/#data>.

Sincerely,

Dr Sasha Dall

Reviewer(s)' Comments to Author:

Referee: 2

Comments to the Author(s).

The initial, background section nicely points out why the dynamics of change in learned song are likely to differ from, and relate to, genetic change, under selection. The paper describes important and original insights, capitalizing on data on territorial/mate-attracting song in a group of birds that can be used to investigate the relationship between learned and evolutionary-genetic changes in song. Looking at these questions with an overarching phylogenetic view, while relating it to the more usual micro-evolutionary perspective, and using a particular example (sunbirds) is mind-expanding for anyone interested in the evolution of social signals, especially those influenced by learning. Given the complexity of the questions, this paper manages to clearly work through the many conceptual complications to an admirable degree.

The one very small clarifying suggestion I have is that it should be pointed out as soon as the word “territorial” is used that the calls that are the focus of this study function as both territorial and mate-attraction signals. Many of the points discussed, such as relevance of song to reproductive isolation and speciation, usually refer to courtship song and female choice, which has different dynamics than do threat and exclusively territorial (e.g., aggressive) signals. [Due to the genetic correlation between signals and choice in courtship signaling the evolution of courtship songs are expected to change more rapidly and attain greater complexity than territorial signals, which are threat signals whose dynamics include imposition of “truth” by ability to back up threats with real strength.] Given the importance of this distinction the place to point this out is on line 106: change that sentence to state that the paper focuses on “learned territorial/mate attracting songs” of sunbirds.

Done.

Small suggestions:

Ln 37 – “dynamics” might be better than “trajectories”

Done (line 37 tracked changes doc, line 37 clean).

40 – longer should be “long”

Done (line 40 tracked changes doc, line 40 clean).

Ln 116 say in a parenthesis what “evolutionary mode” means, e.g. (punctuation, stasis, gradual change)

Done (line 156 of tracked changes doc, lines 129-130 clean).

122 change “of” to for, or regarding

Done (line 160 of tracked changes doc, line 133 clean).

131 – can you change “multifarious” to “not uniform?” Or just variable or “varies in its nature,” which better relates to the next sentence.

We’ve edited this paragraph, and the term multifarious is no longer used here.

Referee: 3

Comments to the Author(s).

The African sunbirds appear to be an excellent system to study song evolution, given the six (or so) species and separated populations within each species, along with the presence of some contact zones. The authors of this manuscript have collected an impressive number of song recordings (419 songs from 142 individuals), setting them up for a reasonably comprehensive survey of song variation within and among the 6 species. This manuscript uses a mapping of song variation onto a molecular phylogeny to conclude that a model with bursts of evolution fits the song variation better than a model of random change at a constant expected rate (Brownian motion). The authors’ interpretation of this result is that songs show a “punctuated evolution” pattern of song variation, and that songs have experience long periods (“> 10⁶ years”) of near stasis.

Unfortunately, this conclusion is not convincing given the analyses and data presented. The main signal for the “burst” of evolution comes from the comparison of the *fuelleborni* species compared to the others. For instance, Fig. 2 shows that the 5 populations of *fuelleborni* have similarly long songs compared to most other species in the phylogeny, and so a comparison of the two models indicates support for the burst model over the Brownian motion model.

We wish to clarify regarding the main signal for pulsed evolution. Indeed, for three of four traits for which the punctuated evolution model is strongly supported, we localize pulses on the branch that subtends *fuelleborni* (log song duration, range of peak frequency, CV peak frequency). However, we also localize one pulse with strong support in the evolution of median peak frequency, and it’s on the branch that leads to the *Uluguru* tip (*loveridgei*), not on the branch that subtends *fuelleborni*. We additionally found some support for pulses in three additional traits, and among these three traits, pulses are localized to two additional branches in the phylogenetic tree. We have added an additional figure (now Figure 2a) to the main text that shows all pulses with moderate or strong support mapped to branches to make this more clear.

There is however an alternative explanation that has not been sufficiently considered by the manuscript: Movement of individuals between the different *fuelleborni* populations could have spread a similar song type between those populations. This could happen as a result either of genetic exchange or purely memetic (transmission of traits through

learning) exchange, and could be a result of selection (a new song type is advantageous) or mutation and drift alone. It is not clear why the manuscript concludes from the similarity of song in these populations that songs have been in stasis for more than a million years. Much of the reasoning appears to be based on the molecular phylogeny suggesting the populations started diverging that long ago, but the genetic data were analyzed with a method that can only present the data in a bifurcating phylogeny, rather than methods that have the capability to indicate genetic or memetic exchange between populations.

We agree with the referee here that movement of individuals among populations, during past periods of time where climatic conditions permitted more connectivity among montane forest-type communities, provides a general mechanism where song phenotypes could be homogenized. We do not think, however, that such broad homogenization across species' ranges could happen without gene flow. We have added verbiage to the end of the introduction (lines 150-153 tracked changes doc, lines 123-126 clean doc) and the Discussion (lines 452 to 464 tracked changes doc, lines 347-359 clean doc) to address this alternative explanation.

We explain here why we do not believe this explanation accords with the data. First, our familiarity with the Eastern Afromontane and Eastern Arc Mountains, and familiarity with the genetic data from this group, has led us to conclude that the widespread movement of individuals among populations with deep genetic structure is not very plausible. We recognize we did not provide this explanation in the prior version, so it has been added in both new sections addressing this idea. Evidence for strong geographical isolation among populations within species is typical of the Eastern Afromontane, and especially the Eastern Arc Mountains, where deep structure among mountain blocks is common not only for forest specialist birds (e.g. Olive Sunbird *Cyanomitra olivacea* Bowie et al. 2004, White-starred Robin *Pogonocichla stellate* Bowie et al. 2006, both species of *Artisornis* tailorbirds Bowie et al 2018) but also for other forest taxa like trees (Jump et al. 2014 Biotropica) and frogs (Lawson 2010). Indeed, "exceptional population genetic structure" is one of the things this region is well known for (Lawson 2013). Such exceptional population genetic structure across diverse montane organisms indicates long-term isolation of sky islands generally.

Specifically for the sunbirds in this manuscript, we do not find evidence for gene flow for distantly related populations *within* named species across this species complex. First, the populations at the tips of long branches in the phylogeny that represent populations within named species in the sunbird species complex (e.g. the population named Njesi within *Cinnyris fuelleborni* and the population named Nguru within *C. moreaui*) have no shared haplotypes with the other populations of those species for mtDNA, and their haplotypes are strongly divergent from the other populations. Thus, for those populations that are critical to making estimates of the duration of stasis, we do not find evidence for gene flow using the data we have. It remains possible that some amount of gene flow could be discovered using more robust sampling across the genome for these taxa, and we plan to perform such analyses in the future. However, given the data that we have, and in the context of sky islands known for long

periods of isolation from evidence in other taxa, we believe we are on firm ground in our interpretation.

Though we do not present these analyses in the manuscript, we used *migrate-n* (Beerli 2001, Beerli and Palczewski 2009) to assess evidence for gene flow in mtDNA between Njesi and Namuli, and between Namuli and the remaining populations within *fuelleborni*. We found that a model with no gene flow is preferred over one with gene flow when comparing models using Bayes factors. Performing this analysis with the available mitochondrial sequence data is limited in that we'd be able to recover gene flow in females only. However, dispersal is female-biased in passerine birds (e.g. Pusey 1987 TREE, Clarke et al 1997 Oikos), suggesting that if substantial gene flow were to occur among geographically isolated populations, it should be detected in mtDNA. Further, we performed analyses for the Mafwemiro and Nguru populations within *C. moreaui*, for which we had multi-locus data. In this case, a model with a gene flow parameter was preferred over one without, however the 95% credible interval for gene flow in the preferred model included 0. Indeed, the 25th percentile of this interval was 0. As such, this cannot be viewed as good evidence for gene flow.

Another reason that stasis is not very convincing is that there is in fact much within-species variation in song. This is mentioned in the text, and is visible in Figure A1. The bivariate plots in that figure show much variation within species, and much overlap between clouds of points for different species. In particular *fuelleborni* shows much variation, and overlap or contact in acoustic space with other species.

We have made several revisions to the manuscript that provided insight into this question. First, as recommended by Referee 3, we performed analyses to estimate the sources of variation among song phenotypes in a hierarchical fashion, using nested random effects in a linear mixed-effect model (LMM), and now present these as part of the results in the main manuscript (lines 221 to 237 and lines 582 to 602 in tracked changes doc, lines 139 to 155 and lines 437 to 444 in clean doc, Tables 1 and A1). In short, there is within-species variation as the Referee points out, and the LMMs show that the variation is minimal *across* populations, *within* species (with the exception of within *mediocris*, where there is strong divergence between northern and southern populations for some traits). The within-species variation is mostly accounted for as within-population variation among individuals in the data set. This analysis corresponds closely with the phylogenetic analyses. The LMM analyses are further detailed below. Secondly, these analyses assume normality of residuals, and as a consequence we performed log-transformation on some variables that we had not log-transformed for the cluster analyses in the prior version. Performing these log-transformations allowed us to recognize that the greater apparent variation in the previous Figure A1 for *fuelleborni* was in part related to the scale of these variables. This was especially true for pause duration, which appeared highly variable within *fuelleborni* in the prior Figure A1, while not being as variable in other taxa. Accordingly, we log-transformed this and several other variables for the multivariate analyses, in addition to the LMMs. We now present a different visualization of the multivariate data in Figure A1. In this version, instead of showing biplots of song measurements themselves, we present biplots of discriminant function scores from a model-based discriminant function

analysis and dimension reduction procedure, with the goal of making the clustering of song phenotypes more visible to reviewers and readers.

We additionally note that our phylogenetic comparative analyses have taken into account within-population variation via the inclusion of standard errors of means for populations, where populations correspond to tips. The analyses show that for traits where pulsed evolution is supported, the variation among populations does not substantially accumulate across populations over time, except at pulses. This is what we interpreted as stasis. We are not claiming that our data set shows that all individuals within species are uniform.

The analysis considers two models of song variation, and it is worth considering whether there are others that may be more consistent with the data and the evolutionary process. A model that was not considered is one in which there is Brownian motion but the rate of that Brownian motion can itself change gradually through the phylogeny. This is likely more consistent with what is known about song evolution. If that is truly how songs are evolving, then the constant-rate Brownian motion model would not be well supported, and the alternative in this manuscript of a burst model might then appear to be favored in comparison. In other words, when only comparing two models, one might look better than the other but they might both be far from the truth.

We understand the concern that both models can be very far from the truth. We can respond to the specific suggestion that a variable-rate BM model with gradually-varying rates could explain the data significantly better, as well as to the general question of whether the pulse model is sufficient for describing the data.

It can be seen heuristically for the song trait referred to as “coefficient of variation in peak frequency” that the variable-rate BM model, with gradually-changing rates, will not fit the data much better. The same argument can be applied to the other traits for which we found strong support for pulsed evolution.

In the following graph at left, each population’s normalized mean trait is represented by a vertical bar; each horizontal bar corresponds to the standard deviation within the population; and each color corresponds to one of the “stasis regimes” our method found, illustrated on the phylogeny on the right. The populations belong either to the stasis regime downstream from the pulse at branch 19 (branch number found below each branch), or the other regime, which includes all other populations.

The traits on the subtree below edge 19 (edge numbers are in red below branches) form a tight cluster, despite the height of this subtree being much more than the length of branch 19 (the numbers in blue are not the lengths, but the pulse support). Similarly, the traits not belonging to this subtree also form a tight cluster, even though the distance between the Mampirwa population and the Mt Kenya population is twice the height of the phylogeny. This suggests a high rate on branch 19 and much smaller rates on all the other edges of the tree. In a variable-rate BM model, this pattern would require at least three different rates: some small initial rate, a drastically higher rate on edge 19, and again a small rate for the descendants of edge 19. Such a fit could not be interpreted as gradual, and would qualitatively simply reproduce the pulse model at the expense of adding one additional parameter. For the phylogenetic tree in this work, which has comparatively few taxa and large error bars in the trait data, a variable-rate BM model and the pulse model would be essentially indistinguishable--except on the basis of parsimony, for which the variable-rate BM model would be worse.

The pulse model, the BM model, the BM model with gradually varying rates, and many other phylogenetic comparative models are all submodels of the multirate BM model with different rates on all the edges. Using this most general model, we found the following maximum likelihood estimates for the rates:

(To do this, we assumed a random rate--within 25% of the BM MLE rate--on each edge of the tree. We then chose a random edge and used gradient descent to minimize the negative log-likelihood by changing the rate on that edge up to 5%, and repeated this process many times, until the rates stopped changing. We obtained the very small error bars by repeating this procedure 20 times. The branch numbers on the x-axis correspond to the red branch numbers in the phylogenetic tree above. As in the manuscript, measurement errors were included in the calculation of the likelihood. This is the same method used in finding optimal branch lengths, see e.g. chapter 23, section "Maximizing the likelihood" of Felsenstein's "Inferring Phylogenies.")

The above figure suggests stasis (BM rate of approximately 0) throughout much of the tree, with one large displacement--just as our pulsed evolution model suggests. More specifically, in the context of this model and the sunbirds' phylogeny, pulsed evolution qualitatively corresponds to comparatively small products *rate \times edge length* on nearly every edge, and comparatively higher products on the edges with pulses. We set the length of each edge in the phylogeny to equal the variance along that edge, i.e. the product of the BM rate on that edge and the length of the time interval corresponding to that edge:

The branches are near-zero for the majority of populations, so these branches cannot be easily seen.

The branch which contributes the vast majority of the variance is the same branch at which our original method detected a pulse. There are also small contributions from the leaves corresponding to Uluguru and Mbulu. As can be seen in figure S9 in the SI, our method found some support for pulses there as well, but rejected them in the model selection stage in order to maximize parsimony. All of this suggests that our pulsed evolution model is sufficient for the sort of data and phylogeny that we have access to.

Due to their length and specificity, we did not include the above explanation in the SI. Instead, we included a short written explanation for why we are using two simple models (SI lines 52--63).

That being said, we agree that including more complicated models would be very beneficial, and we hope to do this in the future once we are able to reliably perform model selection in this context, which we found was not a trivial task.

Although the manuscript is about song evolution, the manuscript is actually not very clear about the pattern of variation in song. It is difficult to see details and compare songs in the Fig. 2C, and showing just a single song for each species seems insufficient given the apparent variation in songs as indicated in Fig. A1. It seems that examining and somehow presenting a summary of song variation within individuals, between individuals within a population, and among populations is important for the reader to understand the overall context of song variation, and only then can hypotheses

regarding the evolutionary process be effectively addressed.

We understand this point and agree with the referee here that it's very helpful to lay out for readers an analysis of the song variation that illustrates how it is structured across levels. We have responded to this suggestion in a previous comment above, also (see that comment above for line numbers and Table numbers). To present such summaries as the referee requests, we have fit mixed-effects models of all the individual songs we analyzed, for each of the 14 focal traits. These analyses are now presented in the main text and Appendix. There are several important aspects of these analyses to point out:

- The marginal R^2 , corresponding to the variance explained by the fixed effect ("species"), varies from .07 to .82 across traits. For those traits with smaller marginal R^2 values (e.g. log median frequency change, log number of elements, coefficient of variation in frequency change), there is little difference among species.
- Generally, there is very little variance explained by variation *among populations within species*. This component of the variance accounts for less than 11% of the random-effects variance. This result corresponds closely to the results we obtain using phylogenetic analyses.
- Both the variance among individuals and the residual variance are comparatively large – i.e. there is variation among songs, but very little of this variation is geographic variation within species.

A few other recommendations:

It would be good if the different figures were more integrated in terms of terminology, use of color and symbols, etc. For example, it is presently unclear whether the mediocris N vs. S in Fig. 1 corresponds to MtKenya vs. Mbulu in Fig. 2. The colors used for taxa in Fig. A1 are different than those used in Fig. 1.

We have altered the figures that show pulse positions on the population tree so that MtKenya is shown as "mediocris N" and Mbulu is shown as "mediocris S" (Figure 2, Figures A4-A9). This better allows comparison between Fig. 1 and these other figures. Secondly, we have produced a new Fig. A1, with colors corresponding to those used in Fig. 1.

There are a variety of phylogenies used in the main paper and the appendix and supplement, and the captions often don't give many details about the source of the data and the method that built the phylogeny. This makes it difficult for the reader to understand the different appearance of the different phylogenies (e.g. compare Fig. A2, where variation among most pops of *fuellborni* seems very low; vs. Fig. 2, where it looks much deeper).

We have added information to the captions of Figures 2 and A2 so that the differences in the trees can be understood in making comparisons between figures.

We also point out that there are long branches within *fuelleborni* in the phylogenetic trees shown in both Fig A2 and Fig 2. In particular, the Njesi population is labeled in Fig. A2, and is approximately as divergent from the remainder of *fuelleborni* samples as *loveridgei* is from *moreaui*.

The paper mostly presents univariate analyses of the different song variables—it says that a multivariate clustering approach generated distinct clusters, but there doesn't seem to be a figure showing those clusters in a way that the reader can evaluate the support for them (e.g., how distinct were they, how much overlap). It would be good if some sort of figure such as a PCA or similar were shown.

In the previous submission, Figure A1 was supposed to show these clusters. We agree that this figure did not do a great job in making it possible to evaluate the clusters. We have replaced Figure A1 with a figure that shows these clusters for the first four discriminant functions from a mixture model-based discriminant function analysis of the multivariate data, together with a dimension-reduction procedure. We think this provides a better visualization for evaluating the clusters than the former Figure A1.

Overall, the authors appear to have a valuable dataset on song variation in this interesting group of birds. A paper that presents the patterns of variation more clearly to the reader and that considers other models than just the two here would be of much value. In particular, it would be good if the authors considered (1) the possibility of the present song variants having spread among populations within a species rather recently, rather than evolving strictly through a bifurcating phylogeny, and (2) the possibility that neither the constant-rate Brownian motion nor the burst model being the best models of real song evolution.

We provided more detailed responses to these issues above. In short, though we cannot rule out entirely the possibility that the homogeneity of song among populations is the result of spread via movement of birds, this mechanism is far less plausible given the available data and the geographical context. Regarding the inclusion of more models, we believe this is a good path for future manuscripts, but because of issues with model comparison/selection, it is outside the scope of this manuscript. We also show above that the Referee's suggested 3rd model (Brownian motion with a gradually varying rate) can be viewed as a generalization of either the pulse model or the single-rate BM model, and that the interpretation of this 3rd model would correspond closely to the pulsed evolution model for those traits where we find strong support for the pulsed evolution model. In other words, we do not think including this model would substantially alter our interpretation.

Referee: 4

Comments to the Author(s).

General comments:

This study introduces a novel technique for measuring pulsed phenotypic evolution, and

applies it to a fascinating empirical system: the evolution of culturally evolving song in an 'land island' species complex. The paper is therefore ambitious, and of potentially broad impact on two fronts.

(1) As someone with only shallow understanding of comparative phylogenetic methods, I found it difficult to assess the new technique for measuring pulsed evolution. It seems to me that the new method could easily have been presented as a standalone paper, and relegating it to a supplemental materials has perhaps meant that less effort has gone into making it easy to follow for interested readers with a limited expertise, like me. I would have appreciated, for example, an explanation of how this approach compares to related (I think) methods examining different evolutionary rates within a phylogenetic tree by O'Meara et al, Revell etc.

We acknowledge that our method should be presented in a stand-alone paper, which we are indeed in the process of writing. We also agree that the SI needs to be more detailed, and have significantly revised it by expanding terse passages, improving the quality of the figures and tables, and introducing some details necessary for the SI to be self-contained. In particular, in addition to minor grammar, syntax and style corrections, we've expanded the general description of the method (lines SI 18--24), used standard language to describe the covariance structure of our model (SI lines 41--50), added a description of how model compares with existing models (lines SI 52--63, 71--79), explicitly described how we normalized the data (SI lines 82--84), explicitly stated why we treat the common ancestor's mean value as a given (lines SI 81--94), made clear that we minimize the negative log-likelihood for a given pulse configuration by including it in the notation (SI Equation 8) and all relevant figures (Figures S2, S7, S8), expanded the description on finding the initial parameter estimates in the pulsed model (SI lines 106--126), expanded the description of the optimization procedure and explicitly stated the parameter count in the pulse model (SI lines 154--160), provided a formula for counting the number of investigated pulse configurations (SI lines 139--151, used in later formulas), and explicitly stated how pulse support on an edge is calculated (SI lines 220--233).

I also think that more could have been done to explain the figures, especially S1, S2, S6 and S7 - even as far as what high values of var_d and var_p imply.

For presentation purposes, and due to the addition of a new figure, the figure labels have changed. We've revised Figure S1 (formerly Figure S2) to present four different covariance matrix shapes, their corresponding likelihood surfaces, and the MLEs obtained using our method for four different pulse configurations. In the caption, we explain that each pulse configuration corresponds to a different likelihood surface and matrix shape, and use the different likelihood surfaces to illustrate the importance of a pulse-specific initial guess for the parameters. For the other figures, we've considerably expanded the captions, improved the image qualities, and labeled the pulse position-dependence in every graph. In particular, we give an interpretation for the size of different parameters.

Furthermore, to make the pulse support clear, we've replaced the phylogenetic trees showing varying levels of support by line weight, and replaced them with a single numbered phylogeny (Figure S3) and per-trait graphs showing pulse support for different edges (Figure S10-S11).

Finally, since this is the introduction of a brand new method in a field where previous studies have been controversial, I think that the paper would have benefitted from efforts to examine its performance with simulated data. I realise that you have simulated data to correct for AICc, but would it make sense to go one step further and demonstrate that this correction is itself reliable, by applying the whole analysis pipeline to simulated data? If given phenotypic data simulated under Brownian evolution, what is the probability that the method (in toto) would erroneously detect pulsed evolution?

and

Lines 171-179. Your statistical analysis did not seem to take into account any correction for multiple comparisons. However, you are comparing 14 song traits (some which are likely to be inter-correlated), without any a priori hypotheses for which should show pulsed evolution. Therefore, it would seem appropriate to me that your statistical discovery of pulses should take into account the risk of an inflated type II error. If you were to simulate 14 phenotypic traits under Brownian phenotypic evolution, what would be the probability of finding 4 show clear signs of pulsed evolution?

(Note: these two questions could be answered in one response, so we put them together)

The fourth referee suggested a more in-depth investigation into the performance of the model selection procedure using additional generated data, which we agree would be an important step in verifying the applicability of the method to other data sets. Because the AICc cutoff and the distribution of $\Delta AICc$ scores changes with both the input trait data and the phylogeny, the performance of the method (as can already be in Figure S4, in which only the parameters are varied, but not the phylogeny) can vary considerably in different input data/phylogeny regimes. Therefore, benchmarking the model selection procedure in general would be quite involved and, in our view, out of the scope of an SI of a manuscript focusing on one specific phylogeny and data set. Applying the results of this proposed study to the work under review would require seeing the performance of the method for data sets and phylogenies similar to the ones used in this work, which is essentially the data used for generating Figure S4 when the phylogeny is fixed. We note that we repeat the entire model selection not just for the phylogeny given in the manuscript, but also for other phylogenies, obtained by bootstrapping (Figure S5). Combined, this is essentially the same data which would have been used to analyze the reliability of this particular study. However, we agree that this is a very good idea, which we plan to pursue in the future. We also appreciate the need for clarity on the performance of a corrected AICc cutoff, and are able to respond to the comments in the context of the sunbirds data in particular.

We chose the AICc cutoff at the 95th percentile of the $\Delta AICc$ scores for the BM-generated data (which was data generated under the BM model, with parameters similar to the MLE

parameters--but not the same). Although the choice of the percentile is suggestive, this correction should only be viewed as making the standard AICc-based model selection procedure more stringent---not as the introduction of a hypothesis test. However, we understand the overall motivation for the question of the probability of erroneously detecting pulsed evolution, and can see the appeal in formulating the analysis in this manuscript in terms of hypothesis testing.

Because the pulse model selection cutoff was set at the 95th percentile of the $\Delta AICc$ values for BM-generated data (with BM parameters only similar to the MLE parameters), the probability of detecting a pulse in BM-generated data is approximately 0.05. Assuming only BM-generated data, the expected number of false positives is $14 \times 0.05 = 0.7$ and the probability of selecting pulsed evolution in at least four data sets (as in our case) is:

$$1 - (0.95)^{14} - \binom{14}{1}(0.95)^{13}0.05 - \dots - \binom{14}{4}(0.95)^{10}(0.05)^4 \leq 0.0042.$$

As can be seen in Figure S4, the distribution of AICc differences can be fairly similar for BM- and pulse-generated data (e.g. in the log duration data) or very dissimilar (as in the CV peak frequency data). This similarity depends on the parameters estimated for each model. Setting the AICc cutoff at the 5% BM-generated AICc difference cutoff can therefore lead to very different false positive rates in different traits. Using the BM-generated data used in figure S3, we've computed the following table of false-positive rates. The traits in which the pulse model was selected are in bold.

Trait	AICc cutoff	False positive rate
CV bandwidth	6.942516	0.098765
CV frequency change	6.464261	0.311688
CV gap after	6.977345	0.072948
CV peak frequency	7.114930	0.059553
Max peak frequency	4.828402	0.188976
Median element duration	6.996302	0.051392
Median gap after	6.264024	0.058537
Median peak frequency	6.268682	0.166667
Min peak frequency	6.357604	0.141176
Range peak frequency	7.483487	0.074534
log elements	4.601177	0.184615
log median bandwidth	4.405359	0.157895
log median frequency change	5.455931	0.086331
log duration	4.905860	0.233010

The probability that all four detected pulses are false positives is the product of the four FP rates in bold, which is less than 0.0002. For the datasets in bold, the actual $\Delta AICc$ values were much greater than the cutoffs, which implies lower FP rates.

An alternative to setting the pulse discovery $\Delta AICc$ cutoff at the 95th percentile of the BM-generated data could have been choosing a cutoff such that the FP rate was the same for all the traits. If we were to do this for an FP rate of, e.g., 0.1, several of the traits with weak pulse

support in the manuscript would have strong support. The choice of an optimal $\Delta A/Cc$ cutoff is complicated and will be explored in future work. We merely point out that the chosen cutoffs appear to be sufficient for the problem at hand.

(2) The paper examines phenotypic evolution of song over a molecular phylogenetic tree of a species complex. I did not find any discussion of potential gene flow between populations - even though this is a closely related group of populations. I think this is potentially problematic for the statistical inference of pulsed phenotypic evolution. Gene flow would, of course, tend to homogenise populations in contact, and give an appearance of comparatively ancient divergence between them and populations not in contact. In other words: gene flow could generate an illusion where a recent selective sweep across populations would appear as rapid evolution before the MRCA. In fact, one example of such gene flow is explicitly mentioned in the Discussion (Lines 258-260)

We understand the referee's point here (one also made by Referee 3), and recognize that this alternate hypothesis was not adequately addressed in the previous manuscript. As a minor point of clarification here, the hybridization mentioned in lines 258-260 in the prior manuscript does not lead to substantial gene flow. The hybrid zone is extremely narrow, and molecular work in a previous study did not reveal substantial introgression at this zone (see McEntee et al. 2016). Instead, it suggested that there was limited, local introgression.

Tests for gene flow should be easy to carry out, and my concern easily dealt with. In this particular case, the situation is potentially more tricky because song might spread between population boundaries even when genes don't - through cultural evolution.

We performed gene flow analyses among populations of *moreaui* and *fuelleborni* using migrate-n. These analyses are described above in this response document. In short, they do not provide any clear evidence of gene flow between the most divergent populations within species. As we agree with Referee 4's point below that the traits we've measured are likely to reflect underlying genetic predispositions and constraints on song, we do not think that widespread transformation of these traits via cultural evolution without gene flow is a plausible mechanism.

An alternative explanation for the pattern of phenotypic divergence that is described is that song is regularly being shared across population boundaries, and that geographic distance is therefore a better predictor of song similarity than phylogenetic distance. It would be great if the authors could test and refute that alternative. I realise that there is some possible evidence that songs may not pass across one population boundary, but can that be applied to the critical boundaries *within* the *fuelleborni* clade, where most of the pulsed evolution seems to occur?

We think we understand this point, but just to clarify, there is support for pulsed evolution across traits in the common ancestor of *fuelleborni*, and little difference among *fuelleborni* populations. This is an interesting suggestion for an additional analysis. In a

manuscript that is already a bit overloaded with analyses, however, we think this is unnecessary to include as we have fairly exhaustively investigated these data to identify where the variation occurs. The LMMs that we have added to the manuscript show that there is little within-species, among-population variation. Given the large geographic distances we've sampled within *fuelleborni*, divergence should be relatively great between e.g. Namuli and Ikokoto, and less between Namuli and Njesi, if geographic distance predicts song similarity. If you look through the Appendix figures A4 to A9, you can see that the values for *fuelleborni* populations are similar among populations, regardless of phylogenetic distances or the (correlated) geographic distances.

(3) At some points, I felt that the manuscript conflated the acoustic measurements that were made with the cultural units that were transmitted from one individual to another. Individual song-types or syllables are learned by birds; in songs like these, they are very high-dimensional trajectories of how frequency changes over time. Measurements such as "mean peak frequency" obviously only capture a small proportion of that variation: I'm sure there are many different song-types in these populations that share the same mean peak frequency. Many of the 14 acoustic features that were measured are inter-correlated, and together, I would bet that they only capture a small proportion of the variation in song structure.

This impinges on the interpretations in the manuscript in several ways. First of all, broad-scale acoustic measurements, like those used here, are likely to reflect general constraints acting on learning (caused by vocal tract morphology, environmental selection, or perceptual predispositions). I would argue that it's the norm in other species that such measurements are stable over large periods of time. For example, studies of several species have explicitly demonstrated that general song features are constant over continent-wide scale populations of millions of individuals (e.g. black-capped chickadees, chaffinches, chiffchaffs, swamp sparrows, just to name a few). This geographic consistency implies considerable stasis over time too.

We agree with the Referee's interpretation of the measurements we've made as being connected to underlying genetic predispositions, and have sought to clarify that this is our view in the manuscript (see lines 80-143 in the tracked changes doc, lines 74-115 in the clean doc). It may indeed be the norm that such measurements are stable over large periods of time (at least one of us thinks this may be the case), but we do not believe that this idea has much if any good empirical support, and there are examples that have found that these types of measurements are not stable even over much shorter periods of time (e.g. over 34 - 37 years in White-crowned Sparrows *Zonotrichia leucophrys* Derryberry 2009 *Am Nat*). There are also examples of birds that have substantial variation in these types of song measurements over their ranges, as in Greenish Warbler *Phylloscopus trochiloides*, mentioned in the main text. Other recently published examples include Hermit Thrushes *Catharus guttatus* (Roach and Phillmore 2017 *Auk*), Common Yellowthroats *Geothlypis trichas* (Bolos 2014 *Auk*), Hermit Warblers *Setophaga occidentalis* (Janes et al. 2017 *J Ornithology*), and *Malurus* fairy-wrens (Yandell et al. 2017). To us, it is evident that there is no established expectation that these dimensions of song should be conserved over long periods of time, and there

are certainly many examples that could be taken to suggest that songs are very labile instead. Indeed, the first sentence of the Yandell et al. 2017 *Malurus* manuscript is “Geographic variation in song is widespread among birds, particularly in species that learn vocalizations.” It might be argued that part of this variation is on the cultural end, but there are many papers that measure variables similar to what we measure and find great geographic variation.

We think that the main difference between studies showing similarity over broad geographic areas (as mentioned by the reviewer, although it was difficult for us to find studies that specifically showed this outside of black-capped chickadees) and our study is that deep genetic divergences, indicating long periods of isolation, occur within the focal sunbird group, and much less so in other potential examples. Continent-wide similarity can be explained by gene flow or very recent geographic expansion in these temperate taxa, but not within the focal group of sunbirds in this study. We have attempted to make this more clear in the new manuscript (lines 145 to 162 in the tracked changes doc, lines 117 to 134 in the clean doc).

So in the Introduction and the Discussion (e.g. L246-249), it feels to me that the manuscript constructs straw men. There should have been clear a priori expectations that song features such as duration and peak frequency show stasis over long periods of time *because* these likely reflect stable constraints on learning, rather than cultural evolution itself, and *because* evidence from other species shows considerable stability over large distances.

We agree that evidence from other bird species shows that traits can maintain similarity across space, and have added in statements in the manuscript to clarify (lines 107-132 in tracked changes doc, lines 101-104 in clean doc). However, evidence of constancy across space does not equate to evidence of constancy across longer durations of time (i.e. >100,000 years), because of the issue of gene flow (as brought up by both Referee 3 & 4) and because of the possibility of recent range expansion from refugia. In these other instances of similarity across space, stability over large distances can be explained by gene flow or recent expansion. Our genetic data show that this is extremely unlikely to be the case in sunbirds, in which populations have accumulated large genetic divergences in mtDNA *within* species.

We reject the idea that the arguments presented in lines 246-249 of the previous ms are straw men, as they correspond to what a lead author expected *a priori* to find before beginning this study, and what reading the literature on the evolution of plastic traits, on sexually/socially selected traits, and on bird song suggests. Also, we are happy to have discovered that our results are intuitive to someone!

(4) Learned bird songs are typically thought to develop through a combination of genetically transmitted constraints (in terms of what is perceived as conspecific, and what can be produced with the vocal organs) and cultural transmission. It is not immediately clear what these results suggest in this regard. It could be the case that the pulses of change represent evolution in underlying constraints. The manuscript is

somewhat unclear about this - it acknowledges that predispositions and constraints exist, but focuses on the learned aspect of song, but then brings up predispositions again in the conclusions. I would personally prefer it if the manuscript was a little more explicit.

We understand this point, especially as when writing the manuscript, we found it challenging to convey this information in a concise way. We agree, of course, that it is likely that changes in the traits we measured are associated with genetic changes. However, some of the traits we measured could almost certainly evolve to some degree without a change in genetic constraints, so at the same time the genetic constraints are unlikely to be “hard” with respect to all measured traits. When tutoring is removed in experiments, the songs of untutored birds differ not necessarily in which memes are used, but also in spectrotemporal characteristics (e.g. Feher et al. 2009 Nature) similar to those we’ve measured. This change is a product of plasticity, and could happen over short timescales in nature. While we believe that there are underlying genetic changes associated with pulses of evolution in these traits, it is still unclear how learning figures into the process of evolutionary change. It could be important. Thus we feel we have to discuss both learning and the genetic constraints. We think the evolution of these traits is itself complicated to discuss, and there is the potential to oversimplify it. However, we have used some of the Referee’s suggested wording to attempt to make these ideas more clear in the manuscript (the two paragraphs starting at line 80 in the tracked changes doc, line 74 in the clean doc).

(5) I think the manuscript neglects literature on learned bird song and speciation. This applies both to modelling studies (by Servedio and colleagues), but also to other studies of island diversification and bird song.

As there is limited space and we are not performing a literature review, it was difficult to include references to everything we would have liked to. Certainly we will miss citing some studies about learned song and speciation. We have included two earlier references to Lachlan and Servedio 2004 in this version (lines 75 and 107 in tracked changes doc, lines 69 and 101 in clean doc). We would be happy to include any particularly relevant study about island diversification and bird song, but are not sure which study or studies this comment is suggesting we cite.

Specific Comments

Line 72 - nb - I don’t think that the reference cited here reports original empirical findings. I am not aware of any strong evidence that novel songs are at a selective advantage (although this has frequently been proposed)

We agree that this idea lacks strong evidence, and as we tried to save some space in the ms, we have removed it.

Line 82 - quite a lot of empirical data suggests an alternative pattern, however: that the “acoustic space” permitted by genetic predispositions becomes saturated within even

moderately sized populations, such that little differentiation is observed between isolated populations. It feels somewhat as if you are setting up a straw man here.

We would be interested in including an example here, but are not aware of the studies referred to, or perhaps have not interpreted them in the same way. It seems like we have drawn from different parts of the literature than Referee 4 would have preferred, but nonetheless we believe we are citing very relevant and influential papers.

Lines 84 - but explicit models of the role of song learning in speciation are not cited. This includes Lachlan & Servedio (which is cited only in the Conclusions), but also other modelling studies by Servedio.

We read through the other modeling papers from Servedio and colleagues, and these others are not as relevant for this paper, as they tend to focus on the evolution of reproductive isolation and less on the way song phenotypes diverge over time.

Lines 92-98 - It seems to me that - as fantastic a study as it is - there is rather too much space taken with discussing the white-throated sparrow study. There are plenty of other studies showing patterns relevant to this one, especially long-term stasis (e.g. Pipek et al in yellowhammers, Lachlan et al. in swamp sparrows).

We have included references to the yellowhammer and swamp sparrow studies in this version (lines 107-132 in tracked changes doc, lines 101-104 in clean doc), and trimmed the part about White-throated Sparrows.

Lines 87-95 - These two examples seem to be confusing two different levels of variation. In greenish warblers, broad scale acoustic features were measured - such as minimum frequency. These measures might be expected to reflect broad constraints on song variation, but not to identify individual song-types.

In contrast, the white-throated sparrow study looks at a particular song-type - albeit with very constrained species levels of variation.

Another key example of pulsed variation is humpback whale song (e.g. Garland et al 2011, Curr. Biol.)

Regarding the two different levels of variation, we agree that the White-throated Sparrow example is more memetic and the Greenish Warbler example more spectrottemporal. But it's not clear to us that these levels are completely distinct from each other, as the incorporation of different memes or song variations can translate to differences in spectrottemporal properties too. Dialect or memetic evolution is often treated as a first step in more extensive or more spectrottemporal divergence, as is written in Pipek et al. 2018. Nonetheless, we hope the other edits we made to the introduction make our points here more clear.

Line 185 - It would be useful to provide a composite tree in the main paper showing the placement of pulses across different traits.

See the new Figure 2, which combines localization of a pulse in a single trait (Fig 2B) with placement of pulses across different traits (Fig 2A).

Line 205. Studies of stasis in learned bird song have been carried out: Pipek et al on yellowhammers; Lachlan et al. 2018 on swamp sparrows, among them.

We have added references to these studies on the Introduction. As we are discussing stasis on a different timescale than these other studies, we do not think it is necessary to include these references at this particular point.

Line 312. I think this is clumsily phrased: there is no empirical whatsoever for genetic assimilation of song traits. The sources mentioned write about genetic assimilation much more generally.

We have altered this sentence to read: "Theory suggests that bird song, as a learned trait, may be prone to evolution via genetic assimilation of phenotypic novelties without an initial genetic basis." (lines 469-470 in tracked changes doc, lines 335-326 in clean doc).

Supplemental Information

At the foot of P6 there must be a typo /missing words in the sentence beginning "Informally..." Please check throughout for typos - in some places I found it hard to follow I think because of similar typos.

Done, and the quality of the text throughout has been edited for clarity.

Table S2 - I found this confusing. Delta A is used both to refer to the four CI columns, but then there is a separate Delta A column, which is not explained.

We've removed the reference to Delta A throughout, edited Tables S1 and S2 to only contain their respective AICc value, and edited Table S3 to clearly outline which Delta AICc values and cutoffs are generated, and which ones come from the fit.

Fig. S5 - the axis of these figures is not described, nor the units.

We've revised the figure (now labeled as Figure S6) to clearly state the units are rescaled, and made a reference to Equation 6, which states the rescaling rule. We've also stated the positions of the pulses in clearer language.

Fig. S8 - the scale bar is not explained. The figure legend should carefully explain what the thick bar means, and what it was based on. After reading the supplemental

materials carefully, I can work most of it out, but the figure legend should make it more explicit.

We agree that the scales were unclear. We felt this figure was difficult to follow, and replaced it with Figures S3 and S9. Figure S3 presents the phylogeny, with units and numbered branches. Figure S9 contains scatter plots which specify the pulse support on each edge. The pulse support is defined in SI Section 4.2 and Equation (21), which is referred to in the caption of Figure S9.

Appendix C

I found the manuscript considerably clearer and more consistent than the previous version - and still of considerable interest. I appreciate the work that has gone into clarifying the methods and interpretation, which has allowed me to understand the paper more clearly than in my initial review.

I have one significant concern remaining. The emphasis of the paper seems very clear now: that although previous work may have argued that bird song evolved in a gradual way, that is not the case here. Instead, songs remain in stasis for long periods of time, and then undergo change in bursts of punctuated evolution.

The evidence for this comes from the contrast of two “idealised” models of evolution: Brownian motion versus punctuated evolution. In the punctuated evolution model, evolution, the variable X_k is set to 0 on all but up to 6 nodes. In the SI, this simplified approach is justified as: “because we are interested in detecting rather than precisely quantifying, punctuated evolution...”.

However, there is a strong emphasis in the paper on the stasis between the punctuated episodes too. I have to admit that I do not really see how evidence *for* punctuated equilibrium precludes gradual evolution between bursts. Because you don't include a model with both Brownian and punctuated evolution possible (as in Landis and Schraiber), I don't see how you can really test the idea of stasis. Surely all you can say is that, for those four traits, your data fits more closely to the punctuated evolution model (with complete stasis in between bursts) than to one with one fixed gradual rate.

I realise that this is closely related to a comment one of the other reviewers made previously, and that you introduced the LMM, partly in response. But the LMM suggests that some of the variation in traits is accounted for between-individual, within-population variance. It might be a relatively low proportion of the evolutionary change occurring in this clade, but (perhaps except for duration) it seems to be clearly >0 . Doesn't that suggest a combination of slow gradual evolution interspersed with rapid punctuated bursts? If so, then the question becomes: is the slow gradual evolution slow enough to justify the conclusion that song traits are in “stasis” for hundreds of thousands of years?

Specifically, I don't see how several statements in the paper can be supported with the methods employed:

LL331-334: what is the evidence that evolution between pulses must be “minimal”?
Authors: We see the reviewer's point here that this doesn't come directly from the relatively strong support for the pulsed evolution model, and instead is evident when looking at the amount of variation in LMMs in context of the phylogenetic models, so we've modified the sentence to say: “Support for the punctuated evolution model across these traits, coupled with the small number of pulses supported for them, is consistent with the hypothesis that evolutionary change is minimal for these traits for long stretches of time, corresponding to hundreds of thousands of years or more on the phylogenetic tree.”

L344: you say that between jumps, evolution must either be in stasis or “highly

bounded". I'm not completely sure what you mean by highly bounded here. Does it allow for gradual evolution?

Authors: We have added the parenthetical statement "(in which evolution is non-accumulating over time)" here.

L406: here again, evolution must be "highly bounded, approaching stasis" suggests that your method allows you to establish an upper limit of the rate of gradual evolution between bursts. If it does, I don't think this has been explained fully.

Authors: We have not placed an upper limit on gradual evolution. Our interpretation relies on the results of the model comparison in addition to visual examination of the data in the context of the phylogenetic tree and the results of the LMMs. Gradual evolution as by Brownian motion would in some cases lead to accumulating changes with increasing phylogenetic distance between pulses. We do not see this in the data for the traits with strong support for pulsed evolution.

L664: "Strongly bounded evolution (stasis or near-stasis)" - same point again. I'm not sure what "near-stasis" really means - in particular in relation to the biological context of the paper. Does near-stasis entail "similar or slower rate of evolution to that found for non-learned traits"? If not, then should your conclusions re song learning be modified somewhat?

Authors: In the models we fit for pulsed evolution, variation exists among populations but does not accumulate. It's not just slower, it's non-accumulating.

In a nutshell: can you rule out the possibility that, compared to a non-learned trait, gradual evolution between bursts is actually rapid, but that it's just much more rapid still during the bursts themselves? If so, that needs to be presented much more clearly to the reader. If not, then I think you need to rethink your interpretation of the results.

Authors: Non-learned traits may as well evolve by punctuated evolution at these shallow phylogenetic levels. As we did not investigate traits not associated with learning in this manuscript, we can say little about them. (Actually we initially also analyzed beak evolution for the same clade, and if memory serves it looks more gradual than song does, but it was too complicated to include those analyses, too.)

Minor points:

L103: I think this is the first use of the term "highly bounded". It would be good to explain exactly what you mean by this. I think you just mean "low rates of evolution". It could simply mean: "with very clearly delimited upper and lower limits of evolutionary rate". But I think that most papers have used bounded evolution to refer to the restriction of the evolution of continuous characters to phenotypic limits. I'm not sure how this applies to your analysis or data, however.

Authors: We have added the term non-accumulating here.

L224: here or in the Table legend, it might be worth explaining that the residual variation refers to between-song-type but within-individual variation (if I've understood correctly).

Authors: added the parenthetical: “(within-individual)”

L227-228: in every species of songbird I can think of, individual-level variation would be much higher than population-level variation. The reason why people have argued that learned bird song might evolve rapidly is because the exceptionally high within-population variation can drive rapid divergence between populations. But that implies a very high level of variation between learned song types within a population that is still often much larger than between-population variation. I don't quite follow the logic for why your results mean that geographic variation between populations must be “minimal”.

Authors: There are almost certainly many other songbirds that would show this pattern, but 1) few named songbird species have such large within-species genetic divergence, making the pattern seen here more ambiguous with respect to evolution over time, and 2) there are others who do not show this pattern, like Greenish Warblers. The inference that the geographic variation is minimal comes from examining sources of variation in the LMMs. The combination of among-song and among-individual variation is far greater than the among-population (geographic) variation.

In addition - I don't think there is any description of how individuals were sampled within populations. Most species of birds show considerable microgeographic variation over a range of a few km. Were individuals sampled across the population range, or mostly from one location?

Authors: This seems to me like it contradicts the point above, at least in relative terms. However, some of the sky island populations are much larger than others. So in some cases, we sampled the entire spatial extent of the sky island (Ikokoto), and in others our recordings were clustered relative to the size of the patch (e.g. Uluguru), but I believe we managed to sample over linear distances of a few km for all.

L352 - typo/missing word here.

Authors: Fixed.

Methods / SI - I do think you need to explain why you didn't use an established method like Landis & Schraiber to test for pulsed evolution. The explanation on line SI79 was not very clear to me in this respect.

Authors: Landis and Schraiber's method requires data associated with many tips on phylogenetic trees. Additionally, we started working on our method before Landis and Schraiber's method came out!

An SI table would be very useful to provide fuller details on sample sizes from different populations, recording locations, sources of recordings, songs analysed per population etc.

Authors: Agreed, though I have run out of time at this point to add in this table. If possible, I would like to add it in.